



# 1 Vista-LA: Mapping methane emitting infrastructure in the Los
# 2 Angeles megacity

Valerie Carranza[1,2], Talha Rafiq[1,3], Isis Frausto-Vicencio[1,4,*], Francesca Hopkins[1,*], Kristal R. Verhulst[1,3],
Preeti Rao[1,**], Riley M. Duren[1], Charles E. Miller[1]
[1] Jet Propulsion Laboratory, California Institute of Technology, Pasadena, CA, 91109, U.S.A.
[2] Institute of the Environment and Sustainability, University of California, Los Angeles, Los Angeles,
CA, 90024, U.S.A.
[3] Joint Institute for Regional Earth System Science and Engineering, University of California, Los
Angeles, CA, 90024, U.S.A.
[4] Department of Chemistry, University of California, Los Angeles, Los Angeles, CA 90024, U.S.A.
* Now at: University of California, Riverside, Riverside, CA, 92521 U.S.A.
**Now at: University of Michigan, Ann Arbor, MI 48109 U.S.A.
*Correspondence to*: Kristal Verhulst (Kristal.R.Verhulst@jpl.nasa.gov)
**Abstract**
Methane is a potent greenhouse gas (GHG) and a critical target of climate mitigation efforts. However,
actionable emission reduction efforts are complicated by large uncertainties in the methane budget at
relevant scales. Here, we present Vista, a Geographic Information System (GIS)-based approach to map
potential methane emissions sources in greater Los Angeles, an area with a dense, complex mixture of
sources.  The goal of this work is to provide a database that, together with atmospheric observations,
improves methane emissions estimates in urban areas with complex infrastructure. We aggregated
methane source location information into three sectors (energy, agriculture, and waste) following the
frameworks used by the State of California GHG Inventory and the IPCC Guidelines for GHG Reporting.
Geospatial modelling was applied to publicly available datasets to precisely geolocate facilities and
infrastructure comprising major anthropogenic methane source sectors. The final database, Vista-Los
Angeles (LA), is presented as maps of infrastructure known or expected to emit methane. Vista-LA



contains over 33,000 features concentrated on <1% of land area in the region. Currently, Vista-LA is used
as a planning and analysis tool for atmospheric measurement surveys of methane sources, particularly for
airborne remote sensing, and methane "hot-spot" detection using regional observations. This study
represents a first step towards developing an accurate, spatially-resolved methane flux estimate for point
sources in California's South Coast Air Basin (SoCAB), with the potential to address discrepancies
between bottom-up and top-down methane emissions accounting.  The final Vista-LA datasets and
associated metadata have been submitted to the Oak Ridge National Laboratory Distributed Active
Archive     Center     for     Biogeochemical     Dynamics     (ORNL     DAAC;
https://doi.org/10.3334/ORNLDAAC/1525).

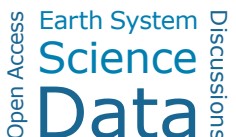

## 1 Introduction

Methane ($CH_4$) is the second most important anthropogenic driver of climate change (Myhre et al., 2013). Recent studies have shown that mitigating $CH_4$ emissions yields large near-term climate benefits due to $CH_4$'s relatively short atmospheric lifetime (Dlugokencky et al., 2011). Reducing $CH_4$ emissions is complicated by the incomplete understanding of the $CH_4$ budget at scales relevant to actionable emissions reduction efforts. Cities are important for GHG mitigation, since they represent high-density emissions regions with the appropriate scale to reduce GHG emissions (Duren and Miller, 2012; Kennedy et al., 2009). Additionally, political will and commitment is needed to implement mitigation efforts for reducing GHG emissions (Gurney et al., 2015). However, enacting emission controls is challenging in urban areas that are highly complex and heterogeneous, with various emission sources located in close proximity.

Understanding urban emissions requires knowledge of source sectors and their respective activities at scales that align with urban policy and planning (typically 10's to 100's of meters). Such information has been assembled for fossil fuel carbon dioxide ($CO_2$) emissions using the Hestia approach, which quantifies urban sources down to the building level (Gurney et al., 2012). To date, Hestia has generated detailed estimates of urban $CO_2$ emissions for several cities, including Los Angeles (LA) (Rao et al., *in review*), Indianapolis (Gurney et al., 2012), and Salt Lake City (Patarasuk et al., 2016). A $CH_4$ emissions product with spatial information equivalent to the scale of Hestia is needed for $CH_4$ emissions mitigation efforts. However, the sources of $CH_4$ differ significantly from those of $CO_2$, which are primarily driven by fossil fuel combustion. Therefore, the methods used to develop Hestia are not directly transferable to $CH_4$, which has distinct source processes and spatial patterns from $CO_2$.

Urban areas are globally significant sources of $CH_4$ emissions, primarily coming from energy use and waste management (Hopkins et al., 2016a; Marcotullio et al., 2013). However, knowledge of the location and relative contribution of these emission sources is highly uncertain, especially in urban areas where energy, waste treatment, and other $CH_4$ emission sources are located in close proximity to one another. Global emissions inventories based on nightlights and/or population scaling methods (e.g., EDGAR v4.2 European Commission Joint Research Centre, 2010; Olivier and Peters, 2005) are limited in their usefulness for estimating emissions at the scale of a city or air basin. Official $CH_4$ emission inventories



made using bottom-up approaches (e.g., IPCC, 2006) underestimate CH₄ emissions and are driven by a
different mixture of sources compared to those inferred from atmospheric measurements, as observed in
Los Angeles (Hopkins et al., 2016b; Hsu et al., 2009; Townsend-Small et al., 2012; Wennberg et al.,
2012; Wong et al., 2016, 2015; Wunch et al., 2009), Boston (McKain et al., 2015), Indianapolis
(Cambaliza et al., 2015), Florence (Gioli et al., 2012), London (Helfter et al., 2016), and San Francisco
(Jeong et al., 2017)). Consequently, there is a need for a new approach of urban CH₄ assessment that
overcomes these shortcomings by incorporating both top-down (observation-based) and bottom-up
(activity-based) information.
Atmospheric CH₄ in the urban landscape is dominated by CH₄ hotspots that primarily come from fossil
fuel-derived sources (e.g., Hopkins et al., 2016b). Many of these hotspots are associated with leaks—
fugitive CH₄ emissions— in natural gas systems (e.g., Jackson et al., 2014; Phillips et al., 2013). Across
the natural gas infrastructure, CH₄ emissions are disproportionately emitted by a small fraction of "super-
emitters" (Brandt et al., 2014). Fugitive CH₄ emissions sources are more challenging to inventory than
activity-related emissions, and contribute to uncertainty in the magnitude and spatial pattern of CH₄
emissions in urban areas (Hopkins et al., 2016b; Lamb et al., 2015). In recent studies, the locations of
fugitive emissions have been identified using observational data, such as mobile surveys, airborne
campaigns, and sustained monitoring (e.g., Cambaliza et al., 2015; Frankenberg et al., 2016; Hopkins et
al., 2016b; Verhulst et al., 2017).
Inaccuracies and coarse information in city-scale inventories of CH₄ pose a direct obstacle to city
mitigation plans. Shortcomings in bottom-up methods have been identified to be: inaccurate
representation of fugitive CH₄ sources at fine spatial scales, the existence of unreported CH₄ sources in
urban areas, and/or incomplete accounting of known CH₄ sources, such as from oil and gas activities
(Hopkins et al., 2016a; Lyon et al., 2015; Zavala-Araiza et al., 2015). CH₄ emissions estimates for urban
regions can be improved by more complete accounting of potential CH₄ sources at the facility scale, along
with targeted observations that can detect fugitive emissions and super-emitter behavior.



Here, we present Vista, a GIS-based $CH_4$ emissions mapping database designed to address shortcomings
in current urban $CH_4$ inventories. Vista encompasses key $CH_4$ emissions categories from the
Intergovernmental Panel on Climate Change (IPCC) GHG Inventory methodology. The primary goal of
this research effort is to improve understanding of $CH_4$ emissions at urban scales with complex mixtures
of sources, exemplified by the LA Megacity. Emissions monitoring and verification efforts in LA are
highly relevant for California's statewide emissions control efforts.  The LA Megacity emits a significant
fraction of California's GHG emissions, with 42% of the state's population concentrated in 4% of the
state's land area (CARB, 2014b).
In this study, we present the Vista-LA database for the spatial domain of California's South Coast Air
Basin (SoCAB), the air basin that contains the majority of LA Megacity GHG emissions. Vista-LA
consists of detailed spatial maps for facilities and infrastructure in the SoCAB that are known or expected
sources of $CH_4$ emissions. Vista-LA illustrates the spatial distribution of potential $CH_4$ sources,
representing a first step towards developing an urban-scale $CH_4$ emissions gridded inventory for the
SoCAB. The final Vista-LA database contains over 33,000 entries, which are presented as $CH_4$ emitting
infrastructure maps. SoCAB is an ideal testbed due to the density of sources and availability of
observations from the LA Megacity Carbon Project (https://megacities.jpl.nasa.gov/portal/) tower
network (Newman et al., 2016; Verhulst et al., 2017), the California Laboratory for Atmospheric Remote
Sensing (CLARS) (Wong et al., 2016, 2015), and a total column carbon observing network site (Wunch
et al., 2009). The Vista data product is a key tool for $CH_4$ emissions research and mitigation efforts; by
(1) mapping areas of $CH_4$ emitting infrastructure, (2) identifying targets for $CH_4$ surveys, and (3) enabling
interpretation of atmospheric observations, including source attribution, and comparison of measured
emissions to permitted or reported emissions. Combined with atmospheric observations, Vista enables
systematic study of urban $CH_4$ emission sources.

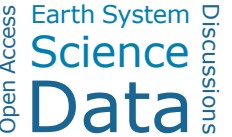

## 2 Methods Overview

### 2.1 Vista-LA Structure and Organization

The spatial domain for the Vista-LA database is SoCAB, the air-shed for the greater Los Angeles urban extent, including the urbanized parts of Los Angeles, Orange, Riverside, and San Bernardino Counties. Vista-LA follows the IPCC $CH_4$ emissions reporting framework (IPCC, 2006). Following IPCC methodology provides compatibility with the State of California $CH_4$ emissions inventory and allows the approach used in this study to be adapted to other regions globally (CARB, 2014a, 2015). For example, Vista-LA can be easily adapted to the Environmental Protection Agency (EPA) national inventory structure since it also follows the 2006 IPCC Guidelines for National GHG Inventories as shown in Figure A1 (EPA, 2016).

The Vista-LA structure enables sectoral tracking of emissions. We used the State of California GHG Inventory for 2015 (CARB, 2016), the most policy-relevant inventory that includes the SoCAB domain, to rank the top $CH_4$ emitting sources (Figure 1). According to the State of California GHG Inventory, ~99% of California's $CH_4$ emissions are expected to result from just three IPCC Level 1 categories – energy, agriculture, and waste, and eight IPCC Level 3 categories – fuel combustion activities related to energy industries and transport (IPCC − 1A1 & 1A3), fugitive emissions related to oil and natural gas (IPCC − 1B2), livestock emissions related to enteric fermentation (IPCC − 3A1) and manure management (IPCC − 3A2); and solid and liquid waste disposal and treatment, including managed waste disposal sites (4A1), and domestic and industrial wastewater treatment and discharge facilities (IPCC − 4D1 & 4D2, respectively) (see Figure 1). The other 27 Level 3 categories cumulatively contribute less than 1% of California's $CH_4$ emissions, and are hence assumed to have negligible impact on SoCAB $CH_4$ emissions. This approach greatly simplifies the database, allowing us to focus our attention on the top-emitting sources. By design, Vista-LA only includes Scope 1 emissions—that is, direct GHG emissions from sources that are owned or controlled by a company within the study domain, as defined by the GHG Protocol (http://www.ghgprotocol.org/corporate-standard). Therefore, sources that are not expected to result in significant direct emissions of $CH_4$ in the SoCAB were excluded, such as emissions from imported electricity, geothermal energy production, and from solid fuels such as coal.



Vista-LA also includes two additional sources that are not explicitly accounted for in the California GHG
Inventory but are potentially significant sources of fugitive $CH_4$ emissions in SoCAB: compressed natural
gas (CNG) fueling stations and liquefied natural gas (LNG) fueling stations, which were categorized
under IPCC Level 2 – 1B2. This case study of Vista-LA focuses on anthropogenic sources of $CH_4$, and
every effort has been taken to make the Vista-LA dataset as complete, accurate, and timely as possible.
Because Vista-LA is designed to incorporate solely anthropogenic sources of $CH_4$, natural $CH_4$ sources
such as wetlands and geologic seeps are excluded this version. This is consistent with the most recent
version of the California GHG Inventory, which categorizes petroleum gas seeps separately as "excluded"
emissions, though they were previously categorized under IPCC – 1B2.
**2.2 Overview of Data Sources**
Within each of the three major $CH_4$ source sectors (IPCC 1 - Energy, IPCC 3 - Agriculture, Forestry, and
other Land Use; and IPCC 4 - Waste), we defined the types of infrastructure associated with emissions.
We sought out publicly available datasets that mapped their spatial locations (Table 1). Spatial datasets
were compiled from reliable and verified public databases on government and federal/state research
agency portals. The data are presented as shapefiles and kmz files that include point, line, and polygon
data. Table 1 summarizes the spatial datasets by the year, source of data, data type (points, lines, or
polygon data) and also indicates the corresponding IPCC Level 3 $CH_4$ emissions category.  Sections 3-5
describe the data sources and information in Vista-LA in further detail and also describe the specific data
processing techniques applied to the GIS dataset for each of the Level 3 emissions category.
Some of the spatial datasets we obtained (see e.g., Southern California Association of Governments,
http://gisdata.scag.ca.gov/Pages/GIS-Library.aspx) and EPA Facility Registry Service (FRS) were useful
for evaluating information from more than one type of $CH_4$-producing infrastructure (e.g., petroleum
refineries and wastewater treatment plants). Due to the variety of data sources used to create Vista-LA,
the same level of detail (e.g., spatial resolution, data completeness, available metadata) was not always
available for every $CH_4$ emitting source. The level of completeness or detail for each spatial dataset will
be discussed below under the data sources and limitations sections. We processed and standardized GIS



datasets through geo-referencing, spatial configuration, and verification using ArcGIS software packages.
All spatial features and raster layers were geo-located using the World Geodetic System 1984 datum and
the Universal Transverse Mercator Zone 11 North coordinate system. Considerations for privacy
including restrictions and limitations on some of these datasets were taken into account for the final
product. Consequently, Vista-LA datasets for natural gas compressor stations and natural gas pipelines
are only included as static representations in Figures 2 and 3. Vista-LA does not include sub-facility level
information. Timely data are critical for understanding methane dynamics in SoCAB, therefore we used
the most current publicly available information in the development of the Vista-LA database.
**3 Energy (IPCC Level 1 – Category 1)**
The Energy (IPCC Level $1 - 1$) sector includes $CH_4$ emitted by fuel combustion activities (IPCC Level 2
$- 1A$) and fugitive emissions from fuels (IPCC Level $2 - 1B$). $CH_4$ emissions from fuel combustion are
mainly produced by energy industries and transportation, with minor contributions from manufacturing,
commercial, industrial, residential and agricultural fuel combustion (CARB, 2016). Fugitive emissions
are defined by the IPCC as an intentional or unintentional release of gas from anthropogenic activities not
including combustion (IPCC, 2001). Fugitive $CH_4$ emissions come from leaks or failures of equipment,
off-gassing, or venting, arising from sources such as natural gas storage facilities, oil and gas wells, and
pipelines. They occur mainly in the oil and gas sector (~95% of California's estimated fugitive $CH_4$
according to CARB, 2016), with a small contribution from industrial and manufacturing sources. Many
facilities, including petroleum refineries and power plants, include both combustion and fugitive $CH_4$
emissions.
**3.1 Fuel Combustion Activities (IPCC Level 2 - 1A)**
Fuel combustion activities (IPCC Category 1A) includes $CH_4$ emissions from energy industries, which
encompass petroleum refining and electricity generation via combustion of natural gas in power plants,
and transportation. Other combustion sources have only a small expected $CH_4$ emission rate (totaling
<0.1% of statewide $CH_4$), according to the California GHG Inventory, hence are not included in Vista-
LA (CARB, 2016). The physical infrastructure associated with combustion and fugitive $CH_4$ emissions



from energy industries in SoCAB are natural gas-fired power plants and petroleum refineries.
Transportation comprises ~1.1% of inventoried statewide $CH_4$, primarily from on-road sources (e.g.,
conventionally fueled cars, light- and heavy-duty trucks), but is not included in this version of Vista-LA
(CARB, 2016).
**3.1.1 Energy Industries (IPCC Level 3 - 1A1)**
**3.1.1.1 Petroleum Refineries (Vista-LA layer)**
*Data sources:*
The Vista-LA petroleum refinery dataset provides location and extent data for 12 facilities in the domain.
The primary spatial datasets for petroleum refineries (IPCC – 1A1) were gathered from the U.S. Energy
Information Administration (EIA) for the year 2016. EIA reports information about all operable
petroleum refineries and electricity generation plants in the United States, including plants that are active,
on standby, and those short-term or long-term out of service (EIA, 2016). Additional information came
from Southern California Association of Governments (SCAG) land use data for the years 2005 and 2012
(see http://gisdata.scag.ca.gov/Pages/GIS-Library.aspx).
*Data processing and validation:*
Petroleum refinery locations were verified using multiple datasets, including EIA, SCAG, and the ESRI
Basemap aerial imagery, and Google Earth imagery.  EIA was the primary source of information, as it
contains the most recent data. SCAG was used to verify that there were no missing petroleum refineries
from EIA. This process provided data quality assurance from the most updated publicly available spatial
database of petroleum refineries.
The original EIA data series on petroleum refineries includes geolocations as points, and information on
production capacity, current and projected capacity of crude oil separated by atmospheric distillation,
downstream charge, as well as fuel, electricity, and steam purchased and consumed by 141 refineries





across the United States (U.S. Energy Information Administration, 2015). This dataset contains
information on nine refineries located in SoCAB, all of which are located in Los Angeles County.
To map the area of petroleum refinery and power plant facilities, we added data from SCAG for the years
2005 and 2012, which maps land use areas to a minimum two acre resolution (see
http://gisdata.scag.ca.gov/Pages/GIS-Library.aspx). The SCAG database only contains land use
classifications for the State of California, and lacks facility-level information. We performed feature
identification using SCAG land use code 1322 "Petroleum Refining and Processing". This category
includes major oil refineries, as well as associated petrochemical plants. This data was used to identify,
extract, and define the spatial extent of each refinery and match the geolocations of the refineries listed
in the EIA 2016 dataset.
The SCAG land use code was used to identify and extract 30 polygons in the SCAG 2005 dataset and 60
polygons in the SCAG 2012 dataset related to petroleum refineries in SoCAB.  Because SCAG polygon
features were fragmented and not assigned to an individual refinery, they had to be manually merged
based on their geolocation and spatial relation to the EIA 2016 dataset.  The polygon features dataset
categorized as "Petroleum Refining and Processing" were merged together and then compared to the nine
refineries identified in the raw EIA dataset.  In some cases, these SCAG polygon features were
geographically misplaced in residential locations or in the middle of streets and had to be manually
adjusted to fit the actual extent of that facility. There were three facilities identified in the SCAG dataset
that were not identified in the EIA dataset. The existence and operation of these three facilities identified
in SCAG were further verified using refinery planning documents and environmental assessment reports
and then were appended to the EIA dataset. The true spatial extent of all polygons was verified using
aerial imagery.  During validation of refinery spatial extents with Google Earth Imagery and Esri Basemap
aerial imagery, focus was given on identifying acres of storage tanks situated in a matrix formation, large
intake pipes, storage vats, and large industrial infrastructure.
*Limitations:*



The additional refineries identified in SCAG and validated through the Vista verification procedures do
not contain facility level metrics that were provided in the EIA dataset. Obtaining detailed sub-facility-
level information for each petroleum refinery will be crucial to developing accurate $CH_4$ emission factor
estimates.
*Results:*
The final Vista-LA petroleum refinery dataset includes all 12 petroleum refineries operated by eight
different companies within SoCAB. The final dataset includes operational data from EIA, recalculated
locational data, and validation notes including any changes made and date of last update.
**3.1.1.2 Power Plants (Vista-LA layer)**
*Data sources:*
The Vista-LA layer for power plants (IPCC – 1A1) relies on data from EIA, SCAG 2005, SCAG 2012,
Google Earth and Esri Basemap aerial imagery (EIA, 2016; see http://gisdata.scag.ca.gov/Pages/GIS-
Library.aspx). The Vista-LA power plant dataset provides accurate location and extent data as well as
facility level information on the type of power generation methods and energy production statistics.
*Data processing, validation and limitations:*
The EIA 2016 contains records for 7,995 power plants in the United States, including 385 power plants
in SoCAB. For our analysis, we selected only the power plants that used the following primary fuels:
biomass, natural gas, petroleum, or other—which matches the methods of the California GHG inventory
(CARB, 2016). This excluded power plants with primary fuel categories such as wind, solar,
hydroelectric, or pumped storage. After filtering by primary fuel type, the new dataset contained 110
power plants in SoCAB.
Polygon features for each of the 110 power plants were created based on Google Earth Imagery, Esri
Basemap Aerial Imagery, SCAG 2005 and SCAG 2012 land use datasets. The SCAG land use code 1431





("Electrical Power Facilities"), was used to verify and determine the spatial extent of the EIA power
plants. SCAG polygons were geolocated with the point data from the EIA dataset. In total, there were
1,490 individual polygon features related to land use code 1431 in SCAG 2005 and 6,932 in SCAG 2012.
In addition to power plants, the SCAG land use code 1431 also includes land used for distribution of
electricity and substations with power plants, hence visual inspection using high-resolution aerial imagery
was required to validate each individual power plant and to generate accurate polygon representations.
When visually inspecting individual power plants, we looked for typical power plant infrastructure
features such as smoke or steam stacks with towers, racks, piping, and vents, transformers and/or electrical
equipment. Some power plant locations were more difficult to validate. In some cases, the power plant
point data was placed on the street near the operating utility and sometimes it did not match the address
that was listed in the metadata. Sometimes the point was located on the center of a site, which could be
within another facility (e.g. a refinery) and thus had to be manually adjusted with appropriate
understanding of the context of its location using Google Earth and Esri Basemap aerial imagery.
Polygons were created using GIS methods including geoprocessing and digitizing with Google Earth and
Esri Basemap aerial imagery as reference. Power plant latitude and longitude coordinates were
recalculated appropriately for each power plant. Power plants whose geolocations were verified but their
spatial extents could not be determined using this method were tagged with a circular placeholder and
their EIA facility level metrics were maintained and marked in the metadata.
We used the 2014 Fossil Fuel Data Assimilation System (FFDAS) point dataset to validate our results
(Asefi-Najafabady et al., 2014). The 105 power plant point locations in the 2014 FFDAS dataset match
with 105/110 power plants in the final Vista-LA layer.  The FFDAS dataset includes only those facilities
registered through CAMD and EIA reporting, which explains the difference in the number of locations
between the two datasets.  One of the five plants is a landfill gas plant, so it is not tracked in FFDAS
because it is not a fossil-based source of $CO_2$ emissions.
We also considered using the 2010 Open-source Data Inventory for Anthropogenic $CO_2$ (ODIAC) for
validation of power plants (Oda and Maksyutov, 2011).  Cross-validation with ODIAC was not
straightforward because the online data product is gridded and is at lower resolution than the EIA and



SCAG datasets. The publicly available version of ODIAC also had significant latency compared to the
EIA and SCAG datasets used in this study.
*Results:*
The Vista-LA power plant dataset merged polygon extent data with the EIA metadata. The final dataset
includes the facility level statistics from EIA and along with data validation information using Google
Earth, SCAG 2005, and SCAG 2012 in the metadata for all 110 power plants originally identified in the
EIA dataset. Based on the 2016 EIA electrical output data, there are only 17 power plants with greater
than 100 Megawatts/hour of electrical output in SoCAB. For this reason, we include the production
metrics in the Vista-LA database, as they may be useful for generating $CH_4$ emission estimates in the
future. The largest producing power plants in SoCAB might be expected to have significant emissions of
$CH_4$ compared to smaller power plants.
**3.2 Fugitive Emissions from Fuels (IPCC Level 2 - 1B)**
Fugitive emissions from fuels (IPCC Category 1B) include $CH_4$ emissions from the lifecycle (production,
processing, storage, transportation) of oil, natural gas, solid fuels, and geothermal energy production
occurring in SoCAB. We omit the latter two sources from consideration since California air quality
restrictions do not permit coal-burning (Perata, 2006), and there are no active coal mining or geothermal
energy sites in the SoCAB. In the California GHG inventory, fugitive emissions are primarily from oil
and gas extraction (30%) and natural gas pipelines (65%) (CARB, 2016). Vista-LA includes spatial
information for oil & gas wells, natural gas pipelines, natural gas storage fields, natural gas processing
plants, and natural gas compressor stations. Petroleum refineries emit fugitive $CH_4$ (IPCC 1B2), but
because of the spatial overlap with refinery combustion emissions at the facility level, we do not treat
them separately in Vista-LA (see refinery layer in IPCC Category 1A1). We also include two more
potentially significant sources of fugitive $CH_4$ emissions in SoCAB that have no assignment in the
California GHG inventory or IPCC categories: compressed natural gas (CNG) fueling stations and
liquefied natural gas (LNG) fueling stations. Vista-LA does not yet include the locations of petroleum
storage tanks due to a lack of publicly available information for these elements. Data for energy-related



sources is also available for purchase from consulting companies; however, one of the objectives of this
work is to generate a product that is openly accessible to the public. Therefore, we did not utilize
proprietary or "for-purchase" information in the development of the Vista-LA database.
**3.2.1 Oil and Natural Gas (IPCC Level 3 - 1B2)**
**3.2.1.1 Compressed Natural Gas (CNG) Fueling Stations (Vista-LA layer)**
*Data sources:*
Geospatial data of active compressed natural gas (CNG) fueling stations was obtained from the U.S.
Department of Energy's (DOE) Alternative Fuels Data Center (AFDC) for the year 2017 (DOE, 2017).
Currently, CNG fueling stations are not included in a separate IPCC category, so for the purposes of this
study we have classified these data under IPCC Level 3 – 1B2)
*Data processing, validation and limitations:*
The raw file was downloaded from DOE/AFDC through a series of queries for compressed natural gas
stations. Next, the CNG dataset was geocoded using latitude/longitude coordinates. Coordinates for this
dataset generated points for 1,792 stations across the United States including 336 in the state of California.
163 of the 336 compressed natural gas stations were in SoCAB, further reduced to 109 after removing
duplicate entries. The geolocations of these 109 CNG fueling stations were verified by comparing the
reported street address to Google aerial imagery (e.g. Google Earth, Google Maps, and Google Earth
Street View) and Esri Basemaps. Out of the 109 points, 88 polygon extent features were created based on
aerial imagery. For the remaining 21 CNG stations, placeholder polygons were created for stations whose
natural gas infrastructure could not be visually identified using aerial imagery, but whose location was
otherwise verified. During visual validation of the CNG stations, we focused on identifying pumps, gas
station infrastructure, and piping/compressed gas storage cylinders near parking lots and salvage yards.
A major portion of the DOE/AFDC dataset placed the locations of CNG fueling stations adjacent to the
fueling station, making it challenging to discern the exact location of station infrastructure on the map.



This dataset counts the entire station as one polygon, despite multiple fuel dispensers. Sub-facility-level
information about individual fueling dispensers is not currently identified in Vista-LA.
*Results:*
The final Vista CNG station layer contains geolocations for 109 polygons. The metadata contains
information about the station name, pressures (units: pounds per square inch or psi), types of dispensing
capability, maximum vehicle size accommodation, and the recalculated latitude and longitude
coordinates, along with validation notes including any changes made and date of last update.
**3.2.1.2 Liquefied Natural Gas (LNG) Fueling Stations (Vista-LA layer)**
*Data sources:*
Similar to CNG stations, fugitive emissions from LNG fueling stations were not inventoried in the 2015
California GHG Inventory. Thus, we assigned LNG stations under IPCC – 1B2. Geospatial data of active
LNG fueling stations was obtained from the U.S. Department of Energy's (DOE) Alternative Fuels Data
Center for the year 2017 (DOE, 2017).
*Data processing, validation and limitations:*
DOE data was originally geocoded using the coordinates listed. The raw DOE dataset contained 187
records of LNG stations across the U.S., 47 of those stations were located in the state of California with
27 currently open and operational in SoCAB. 15 of the 27 LNG stations shared the same location with
CNG stations. The geolocations of the 12 LNG-only stations were verified by comparing the reported
street address to Google aerial imagery and Esri Basemap aerial imagery. Similar to the CNG stations,
extent polygons were generated of the remaining 12 stations using aerial imagery. During visual
validation of the LNG stations, focus was given on identifying gas station infrastructure, and
piping/compressed gas storage cylinders near parking lots and salvage yards.



This dataset is updated annually by the DOE, meaning additional validation will need to be completed in
the future as more LNG fueling stations come online. Two stations were already listed as being planned
in SoCAB and will be operational in less than a year and will need to be added to the dataset in the future.
Similar to the CNG dataset, the LNG dataset assigns the entire station as one polygon, regardless of the
number of fuel dispensers.
*Results:*
The final Vista-LA LNG fueling stations dataset contains polygons for 27 stations in SoCAB. The LNG
dataset also includes metadata describing the station name, pressures (units: psi), types of dispensing
capability, maximum vehicle size accommodation, recalculated GPS coordinates, and validation notes
including any changes made and date of last update.
**3.2.1.3 Natural Gas Compressor Stations (Vista-LA layer)**
The natural gas compressor station (IPCC – 1B2) dataset was obtained using the U.S. Environmental
Protection Agency's Facility Level Information on GHG online reporting Tool (EPA FLIGHT). We
approximated their locations of non-reporting facilities using the EIA compressor station database, which
provides postal-code-level information on compressor stations (Maasakkers et al., 2016). We identified
two natural gas compressor stations in SoCAB. The locations of both natural gas compressor stations
were validated using Google Earth and Esri Basemap aerial imagery (EPA, 2015). Due to restrictions, we
only show the location of the reporting compressor station facility in Figures 2 and 3. The EPA data can
be complemented with the EIA database for non-reporting facilities for potential future development of
gridded emissions products.
**3.2.1.4 Natural Gas Pipelines (Vista-LA layer)**
Information for natural gas pipelines (IPCC – 1B2) was collected from the California Energy Commission
(CEC) and the EIA 2017 dataset (Maasakkers et al., 2016). The CEC dataset provides infrastructure
information of major gas transmission and hazardous liquid transmission pipelines in the United States
for the year 2012 (CEC, 2012). For California, the raw CEC dataset was georeferenced and clipped to fit



the spatial extent of SoCAB. We also obtained high-resolution natural gas transmission pipeline maps
from the National Pipeline Mapping System (NPMS) to validate the CEC pipeline layers. The NPMS
dataset includes a level of detail similar to that of the CEC dataset, but can be obtained for the entire U.S.
There were very minor differences between the CEC and NPMS layers; However, because the NPMS
restricts distribution or visualization of this data, they were retained for internal use only. Due to security
concerns, the CEC dataset is only shown as static representations in Figures 2 & 3.
Unlike the CEC and NPMS data, the EIA dataset is available publicly. The EIA dataset has a lower level
of accuracy compared to the CEC dataset. The positional accuracy of the EIA dataset is ± 3,000 meters
while the positional accuracy of CEC ± 150 meters. The EIA dataset also contains less information on the
disaggregation of pipeline segments. The final Vista-LA natural gas pipeline dataset includes a
georeferenced and processed version of the EIA dataset and contains 111 polyline segments, however due
to the uncertainties noted, it would be ideal to use more spatially resolved datasets for future work.
**3.2.1.6 Natural Gas Storage Fields (Vista-LA layer)**
*Data sources:*
Under IPCC – 1B2, natural gas storage facility point data for the United States was obtained from the
U.S. Energy Information Administration (EIA) online database for 2016 (EIA, 2016). Natural gas storage
geolocations and spatial extent data were obtained using oil field extents from California Department of
Conservation's Division of Oil, Gas, & Geothermal Resources (DOGGR) for 2016 (DOGGR, 2016). The
EPA FLIGHT tool was also used for data quality assurance (EPA, 2015).
*Data processing, validation and limitations:*
Point locations for natural gas storage from EIA in 2016 included 415 points for the entire United States,
three of which are in SoCAB. Since underground natural gas storage in California is done in depleted oil
fields (EIA, 2008), we determined natural gas storage field extents using the 2016 DOGGR dataset, which
contained 516 polygon features for oil field extents in the state of California. Both datasets were first
georeferenced. The EIA metadata contained operation metrics for each storage field, which we appended



to the DOGGR polygon shapefiles for these three extracted entries. The EPA FLIGHT online GHG
reporting tool was used to validate the geolocations of the Aliso Canyon and Honor Rancho storage
facilities. Southern California Gas Company's online information on natural gas storage facilities
validated the geolocation and extent of the Playa Del Rey storage facility.
We did not include former gas storage fields that are no longer used, such as the Montebello Oilfield.
Nevertheless, it is possible these former storage facilities are still leaking, as it takes many years to deplete
the gas to pre-storage conditions (Chilingar and Endres, 2005).
*Results:*
The Vista-LA natural gas storage field polygon layer contains the spatial information and attribute
information of the three natural gas storage facilities located within SoCAB: Aliso Canyon, Honor
Rancho, and Playa Del Rey. The final Vista-LA layer contains metadata relating to field type, company
name, amount of base gas, working capacity, field capacity, maximum delivery, and recalculated
locational coordinates along with validation notes including any changes made and date of last update.
**3.2.1.5 Natural Gas Processing Plants (Vista-LA layer)**
*Data sources:*
Natural gas processing plant (IPCC 1B2) geospatial data was obtained from the U.S. Energy Information
Administration (EIA) online database for the year 2014 (EIA, 2016).
*Data processing, validation and limitations:*
The raw 2014 EIA dataset contained point geolocations for 551 processing plants across the United States,
with six of these in SoCAB, which were georeferenced. Because the raw EIA dataset was limited to a
scale of 1:1,000,000, extensive manual geolocation and validation had to be completed for each plant.
EIA's geolocation was manually augmented using aerial imagery, planning documents and environmental
assessment reports related to the operating companies of each processing plant. Exact spatial extents for



the processing plants were created by identifying infrastructure features such as electrical equipment,
piping, vents, smoke or steam stacks with towers, racks, and transformers using both Google Earth and
Esri Basemap imagery.
*Results:*
The Vista-LA natural gas processing plant layer contains verified geolocated polygons of six facilities
located in SoCAB. The associated metadata includes information on facility and operator name, plant
flow in million cubic feet per day, dry storage in million metric standard cubic feet, energy content of
natural gas in British thermal units, barrels of liquid natural gas stored at each facility, recalculated
locational coordinates of each polygon, along with validation notes including any changes made and date
of last update.
**3.2.1.7 Oil and Gas Wells (Vista-LA Layer)**
*Data sources:*
Data on oil and gas wells was collected from DOGGR for the year 2016 (DOGGR, 2016). The oil and
gas well dataset contains information on well status, type, coordinates, and whether it has undergone
hydraulic stimulation treatment. Another dataset from DOGGR also includes historical production and
injection statistics, owner and operators of the well, and the state of the well.
*Data processing, validation and limitations:*
In SoCAB, the DOGGR dataset includes 32,537 oil and gas wells, associated with activities such as gas
storage, pressure maintenance, water disposal, and other (DOGGR, 2016). Due to the sheer size of the
datasets, we assumed the locations of the wells in the DOGGR dataset to be valid for the purposes of this
study. The dataset includes 5,804 abandoned wells, some of which may be located underneath buildings
and other structures which hinder validation of their locations (Chilingar and Endres, 2005). Validation
of this dataset is beyond the scope of this work, even in cases where manual visual inspection methods





and/or automated feature extraction from aerial imagery might be useful. We discuss possible methods
for automated feature extraction using aerial imagery further in Section 6.
According to DOGGR, well information varies in accuracy, scale, origin and completeness (DOGGR,
2016). DOGGR uses a variety of sources to establish well locations. These sources include handheld
measurements using GPS units derived from DOGGR Division staff, coordinates provided by operators,
well summary reports, official notices regarding the intent to drill, coordinates derived from aerial
imagery, coordinates generated from a tool in MapInfo based on corner call locations, and coordinates
from digitized maps. However, we note that some wells in LA were drilled before accurate records were
kept by DOGGR (Chilingar and Endres, 2005).

## 467  4 Agriculture, Forestry, and Other Land Use (IPCC Level 1 - Category 3)

In the California GHG Inventory, emissions from Livestock (IPCC Category 3A) are the largest of the
IPCC Level 3 categories (Figure 1). Emissions from Biomass Burning (IPCC − 3C1) contribute at the
~0.1% level (Figure 1), and are therefore considered negligible for the purposes of this study. Emissions
from Wetlands (IPCC − 3B4) and all other emissions from the Land (IPCC Category 3B), Aggregate
Sources and Non-CO$_2$ Emissions (IPCC Category 3C) and Other (3D) source types of the Agriculture,
Forestry and Other Land Use category are also considered insignificant in the domain, and were not
included as part of this study. Within the Livestock category, dairies and cattle farms are the major
contributors in the SoCAB region (Viatte et al., 2017). Below we describe our methods for collecting GIS
data related to these activities within the SoCAB.

### 477  4.1 Livestock (IPCC Level 2 - 3A)

The livestock category (IPCC − 3A) includes emissions from enteric fermentation (IPCC − 3A1) and
manure management (IPCC − 3A2). Manure management systems vary from facility to facility and
broadly fall into dry and wet management practices (Kaffka et al., 2016). Manure can be handled and
stored using dry lots, deep pits, solid manure storage, daily spread, digesters (CARB, 2016). In
slurry/liquid systems, waste from feedlots and other livestock areas are washed and is collected in ponds,
which are commonly referred to as anaerobic lagoons (Kaffka et al., 2016). Emissions from manure



depend the type of management practices employed by the farm or facility (Kaffka et al., 2016). Wet
manure management involves washing of feedlots and other livestock areas, and the waste runoff is
typically collected in lagoons where $CH_4$ is produced due to anaerobic conditions. By contrast, dry manure
management practices do not wash waste with water, thus reduce anaerobic conditions. Dairies in SoCAB
primarily use dry manure management practices due to the copious amounts of water needed in wet
manure management practices for the collection, movement and storage of animal wastes. However,
recent mobile measurement campaigns verified $CH_4$ emissions from a small number of dairies with
anaerobic lagoons as recently as 2015 (Hopkins et al., 2016b; Viatte et al., 2017). In addition to dairy
locations, the locations of some anaerobic lagoons were identified as part of the Vista-LA database as
described below.
**4.1.1 Enteric Fermentation (IPCC Level 3 - 3A1)**
**4.1.1.1 Dairies (Vista-LA Layer)**
*Data sources:*
Dairy and cattle farm facility data were collected from the California Regional Water Quality Control
Board (RWQCB), Santa Ana Region. The data was drawn from annual reports, which contain information
on the location of each dairy, the number and type of the herd (i.e., milking cow, dry cow, heifer or calf)
and other livestock located at each facility for the year 2015 (Kashak, 2016). We defined cattle farms as
facilities that did not contain any milking cows. Overall, dairy facilities were primarily found to be
located in the Chino and in the San Jacinto Basins.
*Data processing, validation and limitations:*
First, the raw RWQCB dataset was georeferenced to match the spatial information of the datasets in Vista-
LA. Next, all 110 locations of dairies and cattle farms were validated with Google Earth's historical
imagery tool for the year 2015. Facility addresses and coordinates were used to validate the true locations
of the farms. When verifying with aerial imagery, focus was given to dairy/cattle facility infrastructure
such as: feedlots, manure lagoons, animal housing structures, and open pastures. The dairy and cattle farm



locations were deemed accurate if the geographic location in the RWQCB dataset was confirmed with
aerial imagery and the coordinates did not overlap another facility. During processing and validation, we
identified and manually corrected the locations for two farms with incorrect addresses based on aerial
imagery. Additionally, we corrected geolocations for twelve farms that were located incorrectly in the
original RWQCB dataset using Google Earth aerial imagery and the facility address information given in
the RWQCB dataset.
The RWQCB did not report the quantity of feedlots or manure lagoons per dairy, but did include several
other types of information which will be useful for estimating emissions from manure management,
including: annual manure produced, manure hauled, manure spread to cropland and amount of wastewater
produced. The RWQCB dataset also contained several facilities that were neither dairy nor cattle farm,
such as a livestock market and beef packing facilities. We removed these from the final Vista-LA layer
because we do not expect significant $CH_4$ emissions from these facilities.
It was difficult to obtain spatial extent for each facility because they were difficult to differentiate, as
many facilities displayed common features. For this reason, all final dairy/cattle facility locations are
point-based in the Vista-LA database. The RWQCB database did not report the number of milking cows,
dry cows, heifers or calves for the seven of the farms in Chino. While it does not affect the geolocation
in our facility maps, information on cattle populations specific to these farms will be helpful for estimating
$CH_4$ emissions from these facilities.
*Results:*
The final Vista-LA dairy layer contains a total of 110 livestock facilities in SoCAB: 22 dairies and one
cattle farm in the San Jacinto Basin; and 56 dairies, 26 cattle farms and five other livestock farms in
Chino. Locations for all facilities were validated using Google Earth imagery. Validation notes have been
appended to the spatial dataset for further information on which facilities were corrected for location.



### 4.1.2 Manure Management (IPCC Level 3 - 3A2)

### 4.1.2.1 Anaerobic lagoons (Vista-LA layer)

*Data sources:*

In terms of manure management practices, Vista-LA focused on the collection of GIS structures for wet manure management. Specifically, anaerobic lagoons (IPCC – 3A2) were identified using visual inspection of aerial imagery since publicly available GIS datasets on anaerobic lagoons were not available. Therefore, we created a preliminary geospatial dataset of SoCAB anaerobic lagoons starting from the dairies and cattle farms GIS data. Anaerobic lagoons, also commonly called manure lagoons, are considered sub-facility infrastructures within dairy/cattle farms.

*Data processing, validation and limitations:*

Point locations of manure lagoons at each dairy farm were visually determined using aerial imagery from the National Agriculture Imagery Program (NAIP) and Google Earth's Time Tool for the year 2015. Infrastructure of anaerobic lagoons were identified near a dairy/cattle farm facility by their distinct rectangular shape and brown to dark blue color associated with the color of wash water from manure waste. Often, aerial imagery showed evidence of cattle, further confirming the facility location. Once a lagoon structure was identified, GIS processing tools were used to create point data for the geolocation.

Most anaerobic lagoons in SoCAB were found in the Chino/Ontario region, the area with the densest clusters of dairy farms. However, the identified geolocation of lagoons for year 2015 is likely to change in the near future due to rapid land use developments in the region and fluctuating manure management practices (Hirsch, 2006). Further work could be done with automated feature extraction with contemporaneous imagery, as discussed in Section 6.3.

*Results*



The Vista-LA layer for anaerobic lagoons contains 228 point locations in the Chino, Ontario, and
Riverside regions. The final layer includes geolocations given by latitude and longitude with validation
notes including any changes made and date of last update.
**5 Waste (IPCC Level 1 – Category 4)**
**5.1 Solid Waste Disposal (IPCC Level 2 - 4A)**
Solid waste disposal includes both managed and unmanaged waste disposal sites, as well as uncategorized
disposal sites. The largest $CH_4$ emissions are expected from managed waste disposal sites, hence the site
category we included in the Vista-LA data product.
**5.1.1 Managed Waste Disposal (IPCC Level 3 - 4A1)**
Managed Waste Disposal (IPCC − 4A1) is the third largest IPCC Level 3 emissions source in the
California GHG inventory, trailing only the Enteric Fermentation (IPCC − 3A1) and Manure Management
(IPCC − 3A2) categories. Landfills (solid waste disposal sites) constitute the source of Vista-LA data for
this source. Vista-LA includes both active and inactive landfills, with the status recorded in the metadata.
**5.1.1.1 Landfills (Vista-LA Layer)**
*Data sources:*
The Vista-LA layer for landfills (IPCC − 4A1) was created using the California Air Resource Board's
2014 landfill data and the 2015 California's Department of Resources Recycling and Recovery's
(CalRecycle) Solid Waste Information System (SWIS) dataset (CARB, 2014; CalRecycle, 2015).
Geolocation and spatial extent for each individual landfill facility was generated and verified using the
2005 and the 2012 SCAG land use dataset, Google Earth, and Esri Basemap aerial imagery (see
http://gisdata.scag.ca.gov/Pages/GIS-Library.aspx).
*Data processing, validation and limitations:*



The CARB 2014 landfill dataset contained locational records for 372 potential methane-producing landfills for the state of California with 73 of those landfills located in SoCAB. The CalRecycle/SWIS point dataset contained information on 3,087 landfills for all of California, with 759 in SoCAB. CalRecycle landfills located in SoCAB were queried to isolate only the 353 tagged as "solid waste disposal facilities" or "solid waste landfills". Finally, 19 duplicate entries were identified and removed.

The geolocation and spatial extent of the 334 unique landfills in the CalRecycle/SWIS dataset were verified using SCAG 2005 and SCAG 2012 land use datasets. We used the SCAG land use code 1432 (solid waste disposal facilities) to identify the polygon features associated with active dumps and sanitary landfill operations. We used both SCAG 2005 and SCAG 2012 for maximum amount of information on landfill extent information because neither SCAG dataset contained all landfill locations. In the raw SCAG 2005 dataset, there were 247 individual polygons associated with the land use code 1432 in SoCAB. The SCAG 2005 dataset showed multiple polygon features for the same facility in some cases, so this dataset was further refined by merging multiple polygons that comprise a known facility location based on the refined CalRecycle/SWIS dataset. Simplified polygon features were created for all the distinguishable solid waste disposal sites based on cross-checking the SCAG dataset with the point data from CalRecycle and World Imagery. In Los Angeles County, the 86 total polygons associated with land use code 1432 were aggregated into 18 individual landfills; in Orange County, 60 polygons were aggregated into 7 individual landfills; in Riverside County, 61 polygons were aggregated into 13 individual landfills; and in San Bernardino County, 40 polygons were aggregated into 10 individual landfills. This process was again repeated using the SCAG 2012 dataset that had 211 polygons designated as land use code 1432. Overall, 48 landfills of the total 334 were identified from the intersection of SCAG 2005/2012 solid waste disposal facilities and landfills from the CalRecycle/SWIS dataset.

The location and spatial extent of the remaining 286 landfills from the CalRecycle/SWIS dataset had to be manually validated and/or generated using Google Earth Imagery along with other GIS methods including geoprocessing and digitizing. We were able to verify the location of 188 out of the 286 designated landfills. Unfortunately, their extent and shape could not be determined using imagery since they had long been closed and thus modified or built upon significantly. A placeholder polygon was



generated indicating the historical location with extent and shape determined and digitized using Google
Earth imagery (for example canyons, excavated pits, and barren land). However, land use changes that
occur on surfaces of former landfills vary from site to site. As seen with verification procedures for power
plants, time sensitive imagery is critical when evaluating the existence and geolocations of landfills. Using
the CalRecycle/SWIS metadata for address location for enhanced verification proved to be difficult
because often there were no landfill related features at these addresses, requiring manual geolocation and
correction.
The verified 334 polygons were subset by matching the SWIS number in the CalRecycle metadata to the
SWIS number of the 73 potential methane-producing landfills in the CARB dataset in order to produce
the final Vista-LA dataset.
*Results:*
In total, 73 potential methane-producing landfills were identified, all locations verified, and polygons
were generated in the final dataset using the actual extent or placeholder if extent could not be verified.
The metadata from the CalRecycle/SWIS dataset was appended to the final Vista-LA landfill polygon
dataset. This includes site-specific information, such as throughput, capacity, and waste types
(CalRecycle, 2015). Validation notes include whether facility extent was derived from SCAG 2005,
SCAG 2012, or Google Earth imagery. The final polygon data for landfill extents can be separately
categorized by landfill status: active, closed, clean-closed, closing, absorbed, and inactive. The operation
status breakdown is as follows: 2 were absorbed, 17 were active, 3 were clean-closed (site is considered
to cease to exist as a solid waste disposal site, but records are kept to document the status of the site), 308
were closed, 1 was closing, and 3 were inactive.
**5.2 Wastewater Treatment and Discharge (IPCC Level 2 – 4D)**
Wastewater treatment and discharge includes both domestic and industrial wastewater treatment facilities
(Table 1).



**5.2.1 Domestic and Industrial Wastewater Treatment and Discharge (IPCC Level 3 – 4D1 and 4D2)**


Wastewater treatment in SoCAB is primarily done through aerobic sludge digestion, which has no
associated $CH_4$ emissions in the California GHG inventory (CARB, 2016). However, in low oxygen
conditions, $CH_4$ may be emitted as a by-product of enhanced denitrification present in water recycling
systems (e.g., open tanks in treatment facilities; Townsend-Small et al., 2012). Many wastewater
treatment plants also use anaerobic digesters which collect $CH_4$ for eventual combustion, but may have
fugitive $CH_4$ emissions. Of the urban sources of $CH_4$, this source is perhaps the most uncertain (Hopkins
et al., 2016a). For the purposes of this study, we assume $CH_4$ emissions are most likely to arise from the
plants with the largest daily flow capacity; however, emissions could also potentially arise from various
points of collection and/or drainage of wastewater and sewage.
**5.2.1.1 Wastewater Treatment Plants (Vista-LA Layer)**
*Data sources:*
The final Vista-LA wastewater treatment plant layer (IPCC – 4D1 and 4D2) relies on data from the State
Water Resources Control Board Facility Report Tool (SWRCB FRS), SCAG 2012 land use dataset,
Google Earth and Esri Basemap aerial imagery. The Vista-LA wastewater treatment plant dataset
provides accurate location, extent, and facility level metrics for the largest domestic wastewater treatment
plants in SoCAB.
*Data processing, validation and limitations:*
From the SWRCB FRS, we obtained information for wastewater treatment plants in the LA Basin for
2016, including facility names, addresses, coordinates, and the design flow rate in million gallons per day.
The raw data, which contained information on 152 plants for the state of California, 36 of which were
located within SoCAB, 26 which contained design flow metrics. We geocoded the CARB data as points,
and found many uncertainties in geolocation of wastewater treatment plants. Because of the relatively
small number of plants with metrics in SoCAB, we were able to successfully resolve this uncertainty.





We generated polygons and validated plant geolocation for the 26 SoCAB wastewater treatment plants
using Google Earth and Esri Basemap aerial imagery and SCAG 2012 land use data. SCAG data was
used to first obtain facility spatial extents. The SCAG land use code 1433, ("liquid waste disposal
facilities"), was used to verify locations and determine the spatial extent of the original list of wastewater
treatment plants. In total, 189 polygons were classified with this land use code. 44 of those polygons were
directly matched with point locations from the SWRCB FRS dataset. Since some facilities had multiple
polygons associated to them, they had to be manually merged in order to associate one polygon with one
wastewater treatment facility. After this merging procedure, 44 polygons were merged to 11 polygons
that directly matched the location of 11 SWRCB FRS wastewater treatment plants. Polygons for the other
15 plants were digitized using both Google Earth and Esri Basemap imagery as reference for spatial
extent. Each of the 26 plants was successfully validated using aerial imagery. During the manual
validation procedure, attention was given to identifying features such as spreading grounds, aeration
fields, water injection plants, and circular tanks. EPA FRS data for the year 2013 was used as a verification
source and contains information about wastewater treatment plants operating within petroleum refineries
and power plants (U.S. Environmental Protection Agency, 2016).
The ten additional plants from the SWRCB FRS dataset were not included in the final Vista-LA dataset
because they did not contain facility level metrics and design flow rates, only location information.
Additionally, it is difficult to identify sub-facility plant infrastructure since aerial imagery can only
provide a certain degree of context.
*Results:*
The final Vista-LA wastewater treatment plant layer contains geolocations for 26 wastewater treatment
plant facilities. Five of these wastewater treatment plants were found to be co-located with power plants.
The metadata contains information about the station name, design flow rate metrics, recalculated
locational coordinates of each facility, and validation notes including any changes made and date of last
update.





## 6 Discussion

### 6.1 Vista-LA Data Summary

The Vista-LA database consists of 33,353 individual features as points, lines and polygons among thirteen spatial layers, providing a spatial representation of major $CH_4$ production sources in SoCAB (Figure 2). For the nine polygon layers, Vista-LA depicts the true spatial extent of each facility, a major advance over the original source data. Pipelines are represented as lines, and oil and gas wells are represented as points, which we consider to be the most accurate representation of these sources. The remaining two sources currently represented by points— dairies and anaerobic lagoons— require future work to accurately describe their spatial extents.

The maps in Figures 3-5 show the spatial distributions of potential sources of $CH_4$ in the IPCC Level 1 categories: Energy (1), Agriculture (3), and Waste (4), respectively. The highest density of $CH_4$ emitting infrastructure is located in the western portion of SoCAB in Los Angeles and Orange Counties (Figure 2). The Energy sector, specifically oil and gas wells, account for the majority of the spatial inventory (32,537 features) and are primarily located in southern and northwestern Los Angeles County and northern Orange County (Figure 3). The Agriculture sector is dominated by dairies and cattle farms located in the Chino and San Jacinto Basins of San Bernardino and Riverside Counties (Figure 4). By contrast, landfills and wastewater treatment plants are relatively evenly distributed throughout SoCAB (Figure 5).

In total, Vista-LA polygons cover 117 km$^2$, or 0.68% of the 17,108km$^2$ extent of SoCAB, substantially narrowing the area over which surveys for fugitive and not-inventoried $CH_4$ sources should be carried out. This spatial structure more closely matches the "hotspot" nature of atmospheric $CH_4$ that has been observed in SoCAB at the scale of meters to kilometers (Hopkins et al., 2016b) than is represented by existing gridded products such as CALGEM (10 km x10 km; Jeong et al., 2013) and EPA/Harvard (0.1° x 0.1°; Maasakkers et al., 2016).



## 6.2 Vista-LA Data Completeness and Uncertainty

The goal of Vista-LA is to provide a complete representation of potential anthropogenic $CH_4$ emission sources in SoCAB; however, ensuring complete spatial coverage of important $CH_4$ sources is challenging. We made the simplifying assumption that including the eight IPCC Level 3 sources which constitute ~99% of the expected California GHG inventory emissions would capture the most important $CH_4$ sources in SoCAB, omitting rice cultivation, imported electricity, and coal mining which are not present in the domain (Figure 1). We also added two new source types that are not included in the California GHG inventory: compressed natural gas (CNG) fueling stations (total: 109 locations) and liquefied natural gas (LNG) fueling stations (total: 27 locations). While these sources are not presently included in the inventory, there was sufficient evidence for fugitive $CH_4$ emissions for inclusion in Vista (e.g., Figure 8; Clark et al., 2017; Hopkins et al., 2016b), particularly given that SoCAB contains 32% and 57% of the state's CNG and LNG fueling stations, respectively.

We verified our list of included source categories against the key source categories at the national level from the U.S. EPA inventory (EPA, 2016), and found that rice cultivation and coal mining were the only source types contributing >1% of total emissions that were not included. We also compared Vista's source categories to observations of $CH_4$ hotspots in Los Angeles. The known sources of enhanced $CH_4$ levels— landfills, cattle, water treatment, power plants, CNG fueling, natural gas pipelines, oil refineries, and oil fields— corresponded to Vista layers with the exception of geologic seeps. This correspondence suggests that Vista-LA is well suited for source attribution of anthropogenic $CH_4$ hotspots in SoCAB.

We omitted several categories that might have important contributions to $CH_4$ emissions in SoCAB, such as transportation. Although transportation produces ~1% of California inventoried $CH_4$ emissions (and <0.3% of national emissions; EPA, 2016), it likely comprises a greater fraction of SoCAB emissions given the greater density of traffic in the region. We have chosen not to include a spatial layer for transportation in this version of Vista—we view Vista as primarily a tool for attribution of large fugitive $CH_4$ emission sources, and there is no evidence for this type of emission from conventionally fueled vehicles. Fugitive $CH_4$ emissions detected along roadways are more likely to originate from leaks in



natural gas pipelines that lie underneath the road surface (Chamberlain et al., 2016). Adding a spatial
layer for transportation would be simple to achieve, such as by including a map of the road network or
from an existing high resolution inventory such as Hestia (Rao et al., *in review*), and needs to be included
in an emissions inventory.

We also omitted two possible natural sources of $CH_4$ emissions in SoCAB—geologic seeps and
wetlands—because they are not included in the most recent version of the California GHG Inventory
(geologic seeps are inventoried as "excluded" sources). These sources have not been included in this
version of Vista despite the large observed emissions from geologic seeps in SoCAB (Farrell et al., 2013;
Hopkins et al., 2016b). We have identified possible spatial datasets to include in future versions of Vista
(USGS maps of Natural Oil and Gas Seeps in California, https://walrus.wr.usgs.gov/seeps/ and U.S. Fish
and Wildlife Service National Wetlands Inventory, https://www.fws.gov/wetlands/). There is a potential
that either sources may contribute significantly to the SoCAB emissions budget—we anticipate including
these in future versions.

There is also uncertainty in the spatial representation of sources in Vista-LA. We assumed that the spatial
location of sources by linking facility-level (or pipeline-, or oil well-level) datasets to IPCC source
categories (Table 1), although there may not be a perfect correspondence between mappable infrastructure
and these sources. An alternative approach would have been to map out the lifecycle of each IPCC Level
1 sector (e.g., Energy, Agriculture, Waste), and determine spatial locations of each lifecycle phase in the
domain. We chose the IPCC approach because is more standardized, and hence applicable to other cities.
The lifecycle approach requires local knowledge, and is likely to differ among cities and regions (Hopkins
et al., 2016a). We also assumed that most $CH_4$ emissions come from the main facility associated with an
emitting activity, such as wastewater treatment plants for wastewater emissions, rather than the sewer
network. This assumption may not hold true, for example when manure is exported from dairies and
treated elsewhere. In contrast to waste and agriculture, we included a higher level of detail for the natural
gas system because there is more evidence of quantitatively significant $CH_4$ emissions from distributed
parts of the network (e.g., pipelines).




Finally, we recognize error inherent in the availability of data, and in the original data sources themselves.
Vista-LA relies on publicly available datasets. Consequently, we are constrained by (a) lack of data on
some infrastructure types that may be transient or has never been collected, such as the locations of
manure piles; (b) unavailability of proprietary data, such as the locations of petroleum storage tanks or
gathering pipelines; and (c) data that poses a security risk, and hence cannot be distributed, such as
specific locations of natural gas compressor stations and high pressure transmission pipelines. Within the
datasets themselves, we found errors in geolocation and missing facilities. As described in the text, we
took steps to control these errors inherent to the source datasets by performing visual validation and using
multiple datasets for the same source category. In most original datasets, geolocation of some facilities
was incorrect by up to several kilometers for various reasons, and some geolocations corresponded to the
offices or street address of facilities rather than actual facility location (e.g., for dairies). Raw datasets for
landfills, natural gas processing plants, and wastewater treatment plants also had coarse spatial resolution
resulting in uncertain geolocations, which we corrected. However, we were unable to visually validate all
sources, including oil wells and landfills. With respect to omission of $CH_4$ emitting facilities, in many
cases, the total number of facilities varied among the raw datasets that were used to construct the Vista-
LA layers (e.g., landfills, natural gas processing plants, natural gas storage fields, power plants, and
wastewater treatment plants). For power plants, there are many small facilities in California (around 4,020
plants) that are not included in the EIA dataset (110 plants). Overall, these small power plant facilities
only represent about 3% of the total statewide electricity production related fossil fuel $CO_2$ emissions,
and therefore most are not tracked. Some of these facilities are also not grid-tied, but are facilities that
generate electricity for an industrial process (K. Gurney, *personal communication*). We found that at
least nine power plants were located within the bounds of a petroleum refinery facility in SoCAB. We
have not considered potential emissions from smaller facilities or those contained within other facilities,
but this may be important to consider as part of future work.



## 6.3 Vista-LA Applications

At present, Vista-LA does not contain the bottom-up empirical measurements required for the creation of an accurate fine-scale $CH_4$ inventory (as in Lyon et al., 2015). However, making and interpreting atmospheric $CH_4$ measurements is easier for the dense, heterogeneous landscape of SoCAB with the guidance of the Vista-LA spatial layers. Vista-LA represents a much-needed first step towards the development of a fine-scale urban $CH_4$ emissions inventory that can be used for design and interpretation of $CH_4$ hotspot surveys. Importantly, Vista-LA compliments the tiered remote sensing observation strategies for regional top-down $CH_4$ emission measurements. Figure 6 summarizes potential applications of Vista-LA, which are described in the following sections.

### 6.3.1 Research Planning

Vista-LA is already in use as a planning tool for research aimed toward better understanding $CH_4$ emissions in SoCAB, through guiding design of $CH_4$ super-emitters surveys and appropriate selection of locations for stationary $CH_4$ monitoring. Figure 7 shows how Vista-LA has been used for planning airborne remote sensing campaigns to survey $CH_4$ point sources. Guidance from Vista-LA allows airborne campaigns to be designed for maximum coverage of key infrastructure identified in SoCAB. In Figure 7, aircraft flight lines shown in green illustrate a path optimized for coverage of key $CH_4$ infrastructure in SoCAB.

### 6.3.2 $CH_4$ Hotspot Detection

In addition, Vista-LA can be used to interpret observations of atmospheric $CH_4$ measurements, addressing the challenge of source attribution of $CH_4$ hotspots detected in airborne and ground-based surveys. Pinpointing the source of fugitive $CH_4$ emissions sources in dense mixed-land use urban areas has been an ongoing challenge (Cambaliza et al., 2015; Hopkins et al., 2016b). In many urban areas, many potential $CH_4$ sources are located in close proximity, such as in the Ports of Los Angeles and Long Beach, which contain extensive fossil fuel use and transportation, active oil drilling and refining, and wastewater treatment. Figure 8 shows an overlay of $CH_4$ observations from a mobile $CH_4$ survey in June 2013, where





high frequency in situ $CH_4$ observations were made from a moving van, as described in Hopkins et al.
(2016b). Atmospheric $CH_4$ levels are shown as colored boxes with an "x" in the center, with blue colors
representing near background levels, and warmer colors (red) representing elevated $CH_4$. The same region
was flown by the Hyperspectral Thermal Emission Spectrometer (HyTES) in July 2014. Plumes of $CH_4$
were retrieved from HyTES radiance data, and are shown in green (Hulley et al., 2016). In the scene,
Vista-LA shows roughly a dozen oil and gas wells and a CNG fueling station near the elevated $CH_4$
observations, narrowing down the number of potential sources greatly. Together with wind direction,
these observations plus Vista-LA enable attribution to the facility level.

### 814  6.3.3 Stakeholder and Public Engagement Tool

Vista-LA also has the potential to guide $CH_4$ mitigation efforts by identifying persistent $CH_4$-emitting
infrastructure when combined with atmospheric measurements. In order to control $CH_4$ emissions
effectively in cities, it is essential to understand the fine-scale spatial distribution of stationary sources
from a broad range of industries in the energy, agriculture, and waste sectors. Local regulators and city
planners may be able to use the location information presented in Vista-LA, combined with targeted
surveys or observations, to develop enhanced $CH_4$ monitoring and mitigation strategies for their cities.
Vista-LA may also be extensible to air-quality mitigation efforts. For example, some processes that emit
$CH_4$ also result in co-emission of other gases that are important for climate and air quality. By
incorporating new information in Vista, such as information on permitting and/or sub-facility
infrastructure information, users may be able to evaluate the air quality co-benefits associated with
fugitive $CH_4$ mitigation strategies.

### 826  6.4 Future Directions: Vista $CH_4$ Database

Vista-LA was primarily developed to identify $CH_4$ emitting infrastructure in SoCAB. However, we
anticipate our approach could be scaled to other regions and over larger spatial scales, including the state
of California, the contiguous U.S., and possibly internationally. Expanding the Vista-LA database across
the state of California is highly feasible given that our framework is consistent with how the State of
California reports $CH_4$ emissions. Additionally, many of the raw data sources used in the development of



Vista-LA already encompass state-level or national-level spatial extents (Table 1). In theory, the approach
could also be expanded to any region on Earth, as long as an IPCC (or similar) inventory and geolocation
data for the top-emitting sources are publicly available.  Our framework is also dependent on the
availability of timely, reliable public datasets.  In this regard, Los Angeles and the state of California are
perfect testbeds for development. By contrast, many regions may not have such information available
publicly, especially in developing nations.
Efforts to expand the database could be enhanced by the use of automated feature extraction techniques.
For example, the use of automated feature extraction techniques could expedite the process of identifying,
extracting, and updating relevant infrastructure features. Automated feature extraction involves machine-
learning algorithms used to recognize patterns through image processing (see e.g., Yuan, 2016;
Castelluccio et al., 2015). In this way, aerial imagery in the SoCAB could essentially be used to parse
through and precisely locate features such as converted landfills, oil and gas wellheads, anaerobic lagoons,
and/or wastewater treatment plants. Future work may benefit from the use of automated feature
recognition algorithms using software such as eCognition, ERDAS IMAGINE, GeoMedia, InterIMAGE,
RemoteView and SOCET GXP in order to identify and update spatial information for facilities and
sources not yet known or housed in current publicly available databases.
In the future, the information in the Vista-LA could be used generate a high-resolution bottom-up gridded
$CH_4$ emissions product for SoCAB. While the initial goal of the Vista-LA database was to provide facility
location information, some of the facility- and sub-facility-level activity and operation information
contained within the metadata may also be useful for assigning emission factors to each source (see
Metadata in the Supplementary Information). For example, the Vista-LA dairy layer contains information
on herd population by type, which could be used to estimate emissions factors for enteric fermentation
and manure management. Manure management practices are known to vary widely by region, even within
the state of California. Potentially, the information in the Vista-LA database could also be combined with
top-down observations and provide independent validation of bottom-up $CH_4$ flux estimates, in a similar
approach to that shown in Section 6.2.2. In general, utilizing the spatial information in Vista-LA to assign
$CH_4$ emissions estimates could significantly minimize errors in the spatial representation of sources



compared to previous estimates for this region. Updates such as changes to locations, spatial extents, and
removal or addition of facilities will be required on at least an annual basis to provide timely and accurate
emissions information.

**7 Data availability**

The final Vista-LA datasets and associated metadata are open access and are available in the Oak Ridge
National Laboratory Distributed Active Archive Center for Biogeochemical Dynamics (ORNL DAAC)
(Carranza et al., 2017; https://doi.org/10.3334/ORNLDAAC/1525).

**8 Conclusions**

Vista-LA adopts a GIS-based approach to map $CH_4$ emissions in dense-mixed-land use areas like the
South Coast Air Basin. Characterizing $CH_4$ emissions at the urban scale is incredibly complex, as there
exist thousands of structures known to be associated with $CH_4$ emissions. Vista-LA successfully identifies
33,503 potential $CH_4$ emitters from three IPCC sectors: Energy, Agriculture, and Waste. Vista-LA
contains accurate and validated spatial extent information for nine sources including compressed natural
gas fueling stations, liquefied natural gas fueling stations, landfills, natural gas compressor stations,
natural gas storage fields, natural gas processing plants, petroleum refineries, power plants, and
wastewater treatment plants. It also includes point location of anaerobic lagoons, dairies, and oil and gas
wells as well as a natural gas pipeline network for SoCAB. Vista-LA has been assisted in flight planning
for $CH_4$ airborne campaigns, can be used in potential $CH_4$ hotspot detection, and can essentially be merged
with top-down flux estimates for the identification of individual point sources. In this way, Vista-LA
represents a first step towards developing a gridded emissions spatial product that can illustrate spatial
distribution of $CH_4$ emissions at a fine-scale. By fusing Vista-LA, automated surface feature recognition,
and other various remote sensing point source data products, we could dramatically improve the
attribution of methane emissions.
Vista-LA serves as a prototype resource to aide in the development of high-resolution bottom-up gridded
models of GHG emissions in densely populated urban areas with a complex variety of sources and can
be adapted to larger scales in accordance to characteristics innate to each respective region. The



development of spatially-resolved carbon emission datasets can offer significant advances in
understanding, managing, and mitigating carbon emissions from cities. Generally, uncertainties in
emission sources and their locations in inventories hinder the implementation of mitigation policies. To
be useful, $CH_4$ emissions information is needed at the scale of individual sources. In addition to accurate
and timely spatial information, urban $CH_4$ emissions inventories should also be flexible enough to
incorporate new information, while remaining relevant for observation-based research efforts such as
surveys, hotspot detection, and inversion modeling. Finally, the information should be communicated in
an open-source, transparent, and well-documented manner.
**Acknowledgements**
A portion of this research was carried out at the Jet Propulsion Laboratory, California Institute of
Technology, under contract with the National Aeronautics and Space Administration (NASA). The
NASA DEVELOP program (http:// develop.larc.nasa.gov/) provided the primary support for this project
for VC and TR during 2015. IFV also participated in the NASA DEVELOP program and received support
from the NASA Scholars Program and the Minority University Research and Education (MUREP) Project
during 2014-2016. FMH was supported by an appointment to the NASA Postdoctoral Program at the Jet
Propulsion Laboratory, California Institute of Technology, administered by Universities Space Research
Association under contract with NASA. KRV was supported by the National Institutes of Standards and
Technologies (NIST) Greenhouse Gas and Climate Science Measurements Program. The authors are very
thankful to Dr. Ben Holt, G. Miller, and C. Rains from the NASA/JPL DEVELOP program for helpful
discussions regarding data analysis, and to M. Limb for comments on the manuscript. We are also thankful
for valuable advice and guidance from the California Air Resources Board (ARB) Greenhouse Gas
Emission Inventory Branch, including: A. Huang, L. Hunsaker, K. Eslinger, W. Widger, J. Charrier, and
G. Ruiz, and T. Rose, U. Prins, F. Thong and G. Bemis for assistance with data from the California Energy
Commission, and Edward Kashak from the Regional Water Quality Control Board, Santa Ana Region.
This writing has used information provided by the California Energy Commission. This writing does not
necessarily represent the views of the Energy Commission, its employees, or the State of California. The





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



## Figures

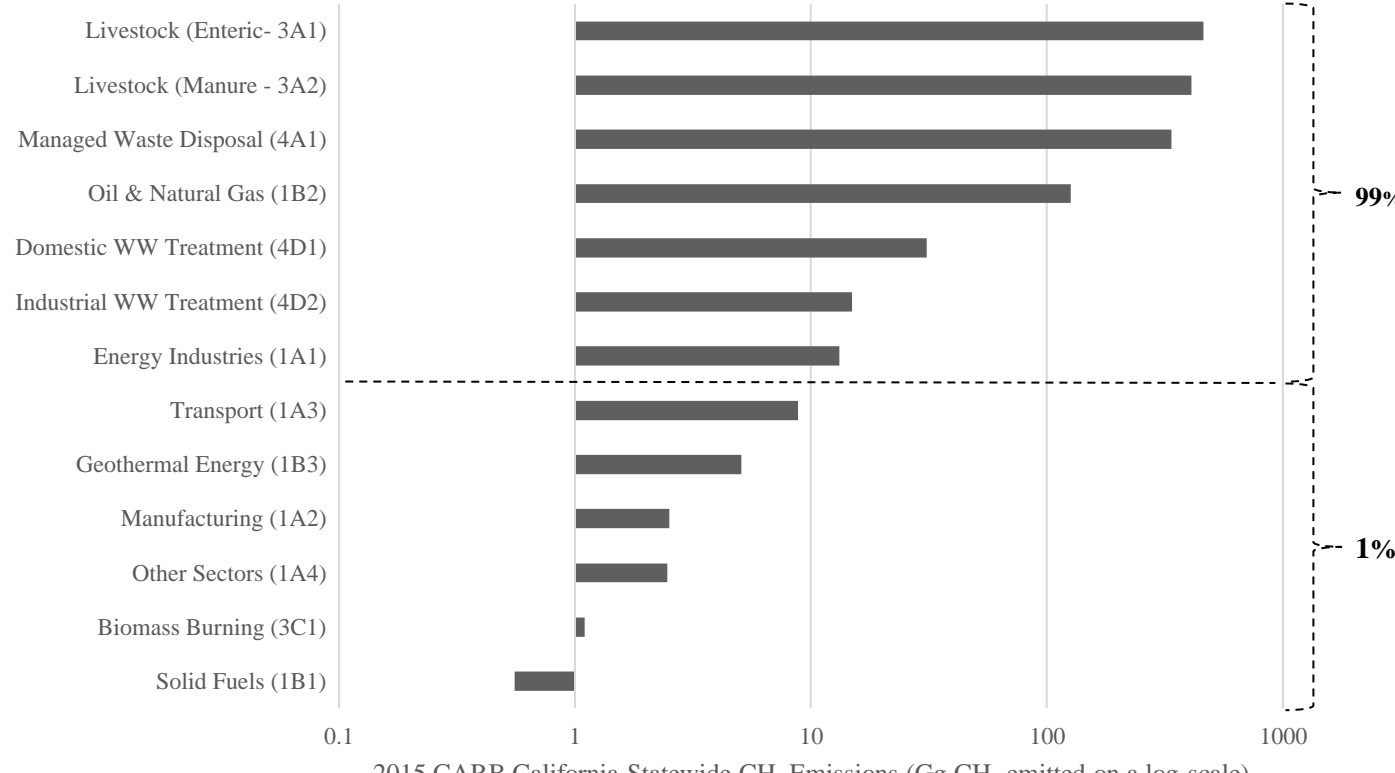

**Figure 1: Ranking of inventoried California CH$_4$ emissions for 2015 by IPCC Level 3 categories.** This graph plots the total emitted Gg of CH$_4$ for each IPCC sector on a logarithmic scale as calculated by CARB for the year 2015. This data is based on IPCC Level 3 categories which are indicated in parenthesis. Only the top 13 of the 35 IPCC Level 3 categories are shown for clarity. The total 2015 emission from these 13 sectors was 1,412.91 Gg CH$_4$. Emissions from activities that are non-existent in the South Coast Air Basin region were not considered part of the Vista-LA database and are not shown in the graph. These activities included imported electricity (IPCC − 1A1), rice cultivation (IPCC − 3C7), and coal mining (IPCC − 1A2). The top seven IPCC Level 3 categories encompass roughly 99% of California's statewide CH$_4$ emissions (~1,392 Gg CH$_4$), are also relevant to SoCAB, and are captured in the Vista-LA database. Note: WW refers to wastewater treatment plants.





**Figure 2: Overview of Vista-LA**. Locations are shown for infrastructure with known or expected potential to emit $CH_4$ in the South Coast Air Basin (SoCAB). Vista layers are categorized by their corresponding IPCC Level 3 from the State of California GHG Inventory (see Table 1). Currently, compressed and liquefied natural gas fueling stations and natural gas storage fields are not included in the statewide GHG emissions inventory, but may be a significant source of fugitive $CH_4$ emissions in the SoCAB (see e.g., Conley et al., 2016; Hopkins et al., 2016b). Note: infrastructure in polygon form is difficult to distinguish from a static zoomed-out image; however, the majority of Vista layers can be viewed at the meter-level. Exceptions to this rule are for natural gas compressor stations and natural gas pipelines due to privacy and security concerns. Total: 33,353 features across 13 layers: 9 polygon layers; 3 point layers; 1 polyline layer.






Figure 3: Energy (IPCC Level 1 - Category 1).


**Figure 3: Energy (IPCC Level 1 - Category 1).** Spatial distribution of infrastructure associated with the energy industry that emit $CH_4$ through fuel combustion (IPCC - 1A1) and/or fugitive emissions (IPCC - 1B2). Natural gas storage fields and compressed and liquefied natural gas fueling stations are currently not explicitly inventoried in the California GHG emissions inventory. Note: infrastructure in polygon form is difficult to distinguish from a static zoomed-out image; however, the majority of Vista layers can be viewed at the sub-meter level. Exceptions to this rule are for natural gas compressor stations and natural gas pipelines due to privacy and security concerns.





**Figure 4: Agriculture, Forestry, and Other Land Use (IPCC Level 1 - Category 3)**. Spatial distribution of dairies and their respective manure lagoons in the South Coast Air Basin (SoCAB), encompassing enteric fermentation and manure management CH₄ sources. The largest clusters of dairies are located in the Chino and San Jacinto regions, with 110 dairies. San Jacinto Basin was home to 22 dairies and one cattle farm and Chino housed 56 dairies, 26 cattle farms and 5 other livestock farms in the year 2015. About 228 anaerobic lagoons were identified in these two clusters.






**Figure 5: Waste (IPCC Level 1 - Category 4)**. Spatial distribution of 73 landfills and 26 wastewater treatment plants in the
South Coast Air Basin (SoCAB).



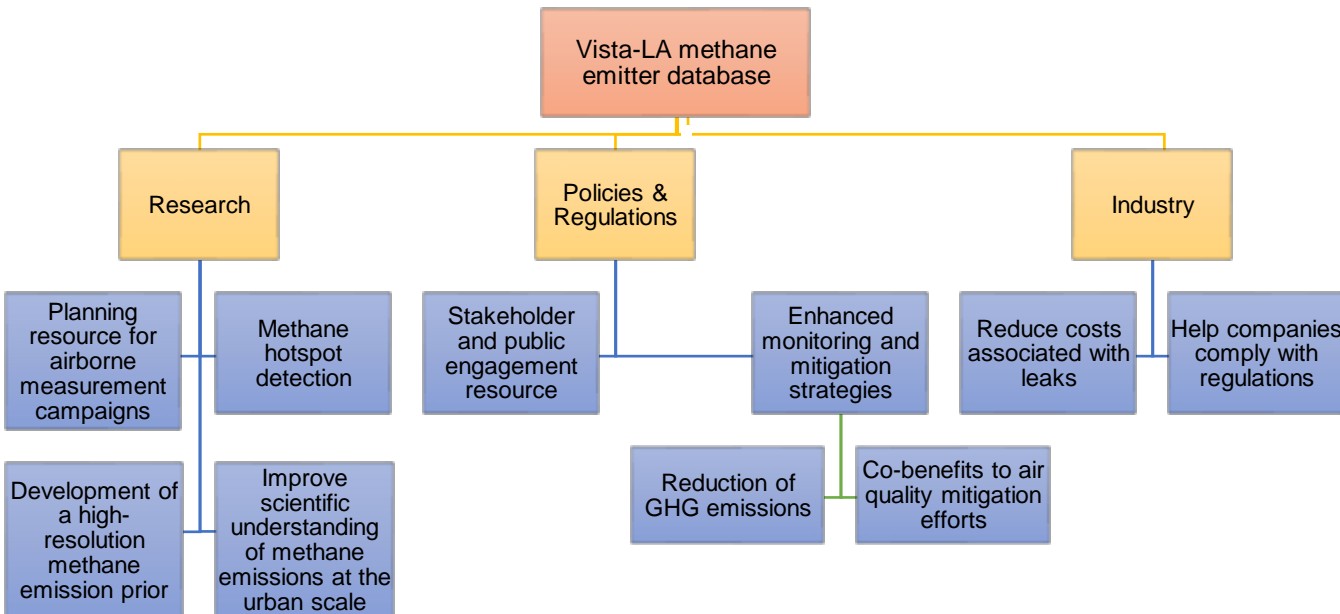

**Figure 6: Overview of applications of the Vista-LA CH₄ emissions mapping tool.** Vista-LA can provide numerous
applications and benefits to research, policy, and industry.





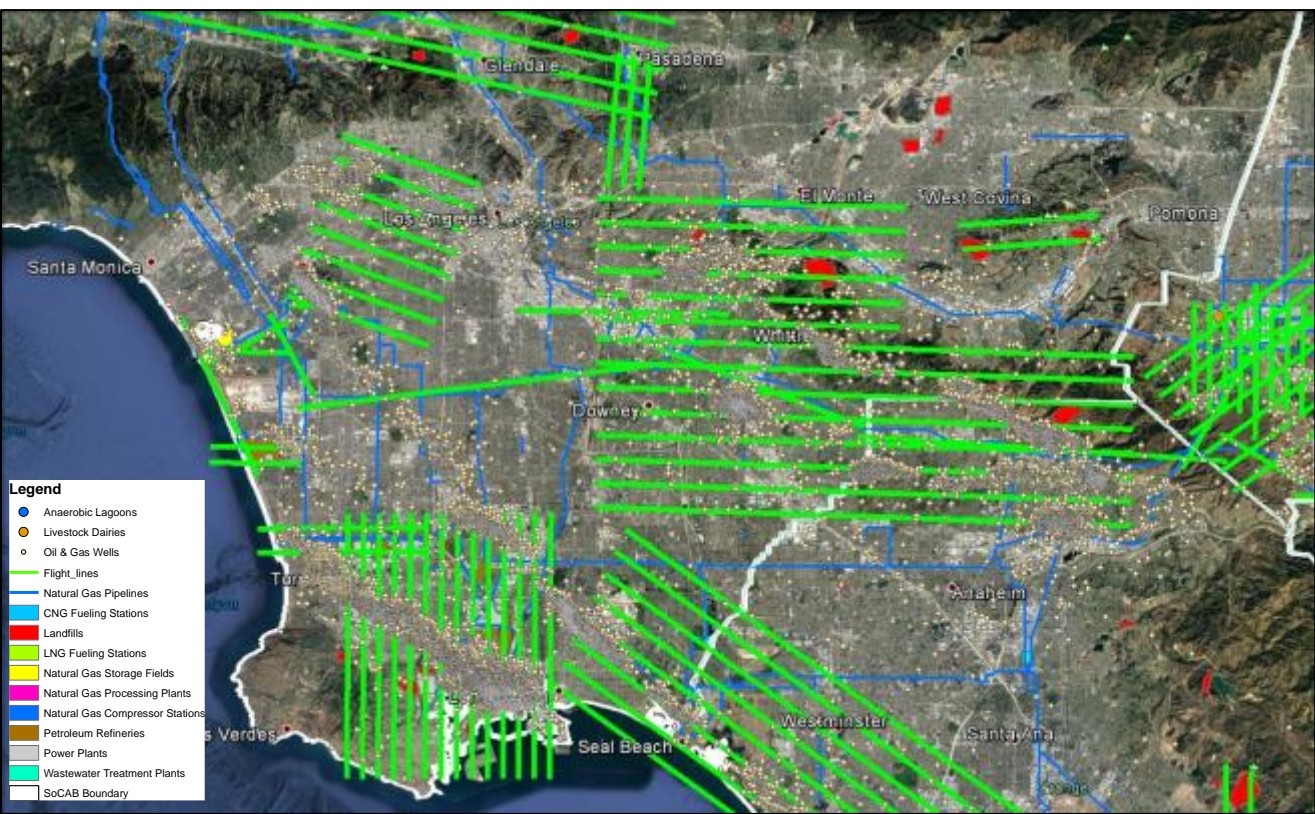

**Figure 7: Vista-LA as a flight planning resource**. The flight path of an airborne CH$_4$ remote sensing campaign to survey
CH$_4$ point source emissions, shown as green lines, was optimized to include CH$_4$ emitting infrastructure for key sources shown
in Vista-LA.



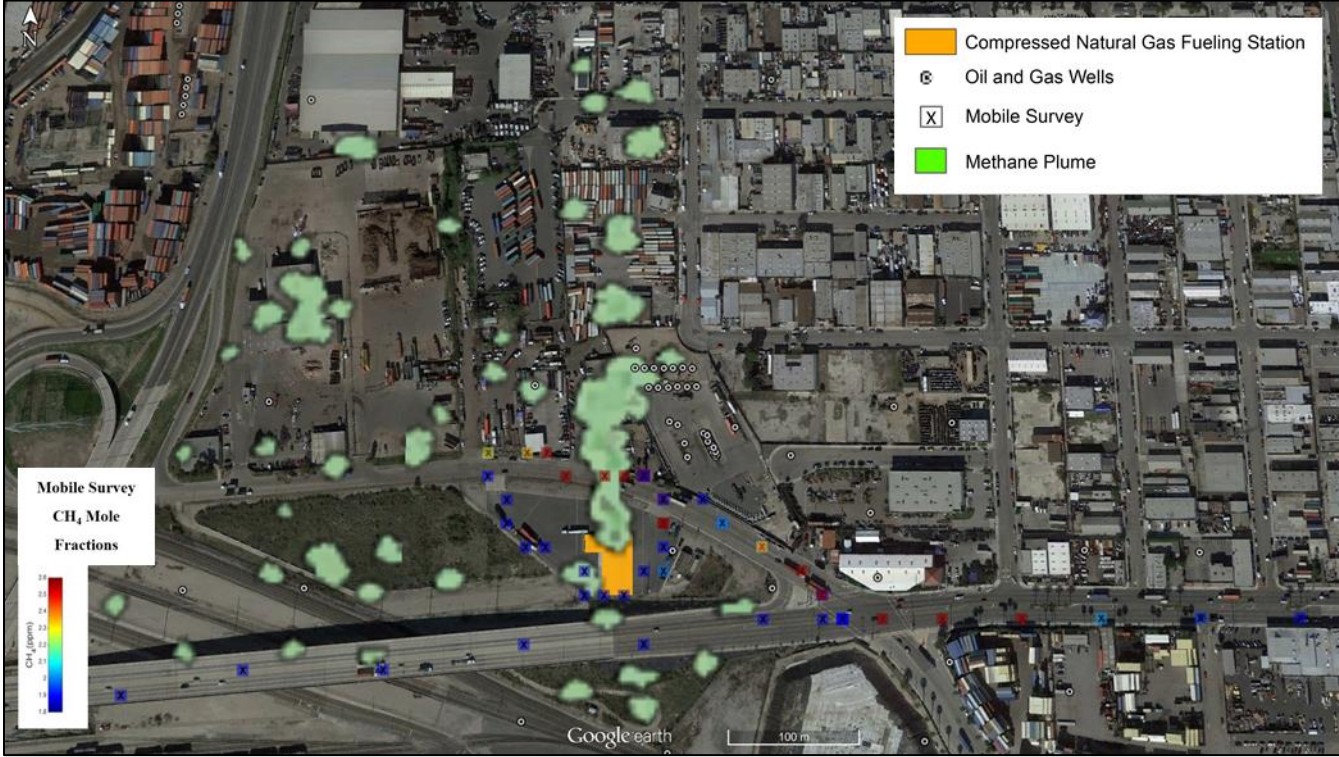


**Figure 8: Application of Vista as a source attribution tool for CH₄ hotspots**. Vista-LA can be used to detect and determine the source of CH₄ hotspots from on-road mobile surveys and aircraft measurements. In this example, Vista-LA layers are shown with CH₄ plumes from airborne imaging by the HyTES instrument and atmospheric CH₄ levels from a mobile survey in the Port of Long Beach, California. The co-location of the green colored CH₄ plume from HyTES (July 2014) and the red point observations of enhanced CH₄ levels from the mobile survey (June 2013) suggest that the CNG fueling station, shown as an orange polygon, is the source of observed CH₄ emissions.



## Tables

**Table 1: Summary of Vista-LA layer**s. Vista-LA layers, representing $CH_4$ sources corresponding to IPCC Level 3, are shown organized by IPCC greenhouse gas emission reporting taxonomy. The source and year of the raw datasets, the maximum spatial coverage, number of features and format are also given for each Vista-LA layer.

| CH₄ Sector | CH₄ Source Type | Vista-LA Layers (CH₄ Source) | Data Source (Year) | Raw Data Spatial Coverage (Data Source) | Vista-LA No. of Features | Vista-LA Data Format |
|---|---|---|---|---|---|---|
| **IPCC Level 1** | **IPCC Level 2** | **IPCC Level 3** | | | | |
| **1. Energy** | | **Energy Industries (IPCC - 1A1)** | | | | |
| | **1A Fuel Combustion Activities** | Petroleum Refineries[a] | EIA (2016) SCAG (2005, 2012) | CONUS (EIA) California (SCAG 2005, SCAG 2012) | 12 | polygons / kmz |
| | | Power Plants[a] | EIA (2016) SCAG (2005, 2012) | CONUS (EIA) California (SCAG 2005, SCAG 2012) | 109 | polygons / kmz |
| | | **Oil and Natural Gas (IPCC - 1B2)** | | | | |
| | **1B Fugitive Emissions From Fuels** | Compressed Natural Gas (CNG) Fueling Stations[b] | U.S. DOE AFDC (2017) | CONUS | 109 | polygons / kmz |
| | | Liquefied Natural Gas (LNG) Fueling Stations[b] | U.S. DOE AFDC (2017) | CONUS | 27 | polygons / kmz |
| | | Natural Gas Compressor Stations[c] | EPA FLIGHT Tool (2016) | CONUS | 1[c] | polygons / kmz |
| | | Natural Gas Pipelines[d] | CEC (2012)[d] EIA (2017) | California (CEC) CONUS (EIA) | N/A 111 | N/A polylines / kmz |
| | | Natural Gas Processing Plants | EIA (2014) | CONUS | 6 | polygons / kmz |
| | | Natural Gas Storage Fields | DOGGR (2016) EIA (2016) | California (DOGGR) CONUS (EIA) | 3 | polygons / kmz |
| | | Oil and Gas Wells | DOGGR (2016) | California | 32,537 | points / kmz |
| **3. Agriculture, Forestry & Other Land Use** | **3A Livestock** | **Enteric Fermentation (IPCC - 3A1)** | | | | |
| | | Dairies | RWQCB (2015) | Chino, Ontario, Riverside Areas | 110 | points / kmz |
| | | **Manure Management (IPCC - 3A2)** | | | | |
| | | Anaerobic Lagoons | NASA JPL-Caltech\RWQCB (2015) | Chino, Ontario, Riverside Areas | 228 | points / kmz |
| **4. Waste** | **4A Solid Waste Disposal** | **Managed Waste Disposal (IPCC - 4A1)** | | | | |
| | | Landfills | CARB (2014) CalRecycle (2015) SCAG (2005, 2012) | California (CARB) California (CalRecycle) California (SCAG 2005, SCAG 2012) | 73 | polygons / kmz |
| | **4D Wastewater Treatment & Discharge** | **Domestic and Industrial Water Treatment & Discharge (IPCC - 4D1 and 4D2)** | | | | |
| | | Wastewater Treatment Plants | CARB (2016) SCAG (2005, 2012) | California (CARB) California (SCAG 2005, SCAG 2012) | 26 | polygons / kmz |

[a]Sources may also include fugitive emissions that fall under IPCC source type 1B
[b]Source not currently included in the California Air Resources Board's 2010-2015 GHG Inventory.
[c]Only includes reporting facilities
[d]CEC pipeline data only available as a static representation in Figures 2 and 3.



**NOTE:**
CalRecycle = California Department of Resources Recycling and Recovery
CARB = California Air Resources Board
CEC = California Energy Commission
CONUS = Contiguous United States Region
DOE = U.S. Department of Energy
DOGGR = California Department of Conservation, Division of Oil, Gas, and Geothermal Resources
EIA = U.S. Energy Information Administration
EPA FLIGHT Tool= U.S. Environmental Protection Agency Facility Level Information on GHG Tool
EPA FRS = U.S. Environmental Protection Agency Facility Registry Service
NPMS = National Pipeline Mapping System
RWQCB = California EPA Regional Water Quality Control Board, Santa Ana Region
SCAG = Southern California Association of Governments



## Appendix

**Figure A1: Comparison between GHG emissions inventory reporting structure for the State of California vs. the United States.** Vista-LA complies with the State of California's GHG emissions inventory structure, but can be adapted to different regions, such as for the national GHG emissions inventory of the United States. Arrows indicate links between sector levels of the two GHG inventories.

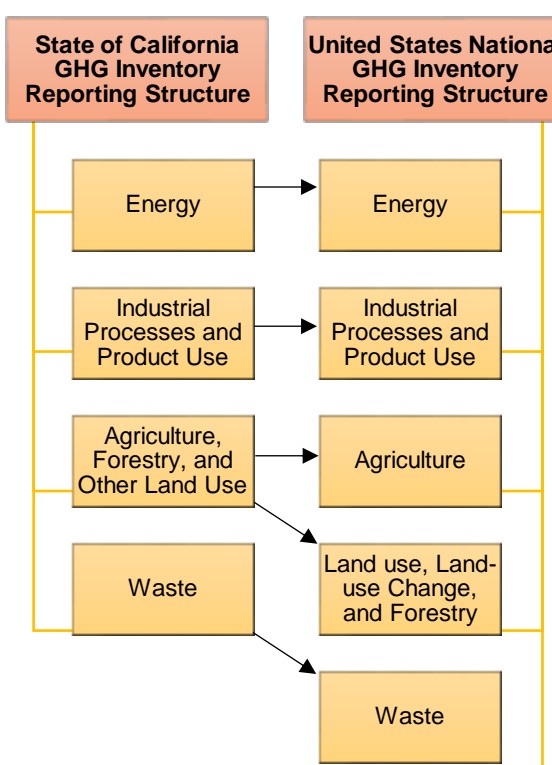