# Peer review of "Vista-LA Data Information"

_Earth System Science Data, 2017_

## Referee Comment (RC1) · Anonymous Referee #1 · 21 Aug 2017

1. In general the manuscript describes an important GIS dataset for assisting with the improvement of methane emission estimates in the South Coast air basin. The authors make a strong case for why the methods described will be useful in other areas, especially in California, but even nationally and internationally. At this stage, the dataset does not include emissions themselves; but, the attributes data do include some activity information that will be useful in making bottom-up emission estimates in the future. A table that summarizes, for each major source type, the level of completeness of the activity data included in the dataset would be a useful addition to this manuscript. Are all, most, some, or none of the needed activity data included in the dataset for each of the major source categories included?

2. Lines 39-40: "Recent studies have shown that mitigating CH4 emissions yields large

near-term climate benefits due to CH4's relatively short atmospheric lifetime (Dlugo-kencky et al., 2011)." Suggested clarification: Methane yields large near-term benefits do to its short lifetime AND its high GWP.

3. Several of the statements in the Introduction section, added to support the utility of the dataset, appear to be in conflict:

a) Urban areas are globally significant sources of methane (line 57).

b) Urban methane is mostly from fossil fuel sources (line 72).

c) By far, most methane in California is from livestock and waste (Figure 1).

Please clarify the text in the Introduction section to explain how statements a) and b) do not conflict with information presented in Figure 1.

4. In the Introduction section, it would be helpful to mention the CALGEM dataset where other relevant datasets are discussed since CALGEM is discussed in the Discussion section (line 698).

5. Lines 714-715: "...rice cultivation and coal mining were the only source types contributing >1% of total emissions that were not included." Statements here were confusing to me. Is the "total" referred here the US total? Please clarify.

6. Line 206 and line 239 mark subsections "Data processing and validation:" and "Limitations." It is unclear why in other parts of the manuscript these subsections are combined. Recommend: Combine these subsections or separate the equivalent subsections to maintain parallel structure with the manuscript.

7. Line 325: "...these data under IPCC Level 3-1B2)" seems to be missing text.

8. Line 493: "data was" → "data were".

---

## Referee Comment (RC2) · Anonymous Referee #2 · 27 Sep 2017

General comments: The work presented here identifies potential sources of methane emission in South Coast Air Basin of California. I believe this is the first of its kind reported in terms of details on the locations of individual facilities and additional information attached to the spatial information. I think that the authors did a tremendous amount of work and the work deserves publication after addressing the comments.

My main comments are: 1) I appreciate detailed information on how to process the data (e.g., geo-referencing) but the paper is too long. The information is useful and I recommend the authors move some of them to the supplemental. 2) I would suggest that the authors clearly state why the spatial extent (i.e., polygon delineation) is useful. As commented below, I am not convinced that the polygon is better than a simple representation as a point. This work will be useful for inversions but I don't see enough

benefits from polygons, except for a few sectors such as dairy farms. Polygon work is associated with errors and makes it difficult for the authors to update the database in the future; the authors already used a significant amount of time in manual work. 3) I realized that this work is not about mapping "emissions" when I expect some results on emission mapping. So the authors need to clarify this at the beginning. The results section is somewhat boring because it just lists datasets, not something meaningful to be absorbed. 4) There are many figures, but the authors do not link the text with figures, only providing a minimum description about the figures.

Please address the detailed comments below.

Detailed comments:

L20: essentially SoCAB?

L33: globally or California or somewhere else?

L41: very vague. "at scales relevant to actionable emissions reduction efforts", Which scale?

L48: any reference for the urban policy scale? Why the entire city? In SoCAB, there are so many small cities. Within those small cities, policies are different from meter scales? Activities vary significantly place to place, but I don't believe such a large variation in urban policies, in particular for GHG control.

L52 – 54: This needs more explanations. Why do we need such a fine-scale emission map? Because the sources of urban $CH_4$ are mostly co-located with those of $CO_2$? Here, we need to be clear. For simple inventory purposes, we may need really fine-scale maps. But as a prior model in atmospheric inversion, do we really need meter-scale maps when the transport model cannot really resolve at such a fine scale? Please add more comments. Otherwise, it sounds like "we just developed a fine-scale product because it is better."

L54 – 56: agreed.

L58: we know the location of a lot of sources for urban areas relative to rural areas. What we don't know is the activity levels and emissions.

L63: reference? I think at least Wunch et al., 2009 mentioned this limitation.

L72 – 73: What about landfill. In SoCAB, fossil-fuel sources cannot necessarily be "dominant" given the uncertainty of recent top-down studies in the region. Furthermore, depending on the definition of the hot spot, hot spots from fossil sources alone cannot be the dominant source in SoCAB. I feel that the authors touch lots of things related to urban CH4 emissions, but I am not sure that the authors are making effective arguments towards the goal of this new study. Literature review is not only scattered here and there but also somewhat inaccurate. I would recommend that the authors revise the introduction section with more clarity and accurate statements.

L89: Did the authors clearly lay out the shortcomings of current urban inventories. Which urban inventory? Here is the problem. The authors try to deal with the "general" urban CH4. Please focus on SoCAB. The authors don't even mention which inventory is available for SoCAB. There are several spatially explicit emission maps that include California and SoCAB. What are the shortcomings of those? The authors should spare the introduction section in describing weaknesses with current inventories (CARB, EDGAR, EPA spatial (ES&T, 2016), CALGEM, etc.) and the contribution this work can make over them.

L93: For those outside California, the use of LA Megacity is confusing since the authors introduced SoCAB. For inventory purposes, SoCAB should be preferable because CARB uses this air basin for regulatory purposes.

L100: What are potential source vs. facilities/infrastructure sources?

L193: I am curious about the portion of transportation in SoCAB. It would be useful to look at the transportation emission for SoCAB from Maasakkers et. al. 2016. Although the portion of transportation relative to the total may be small, transportation seems to

be important in this high-resolution maps, in particular in this highly urbanized region.

L212: I wonder if EIA data for refinery provide spatial information (i.e., lon/lat), probably only zip code. If only zip code, then this sentence is not really correct. EPA mandatory reporting (GHGRP) may provide exact locations.

L217 – 224: It should be useful to check with EPA's mandatory GHG reporting system.

L228: Please confirm that the EIA dataset has exact point locations. Also, cross-checking with Facility Registry Service (FRS) should be useful

L252: reason for claiming "The Vista-LA power plant dataset provides accurate location and extent data"?

L372: The authors may want to verify the compressor stations in SoCAB comparing with those from the California Energy Commission.

L396: Why did the authors use the EIA pipelines instead of those of CEC? Due to security concerns although the authors do not state it (only for NPMS).

L410: Don't need repeat this geo-referencing unless there was a need to manually geolocate facilities, e.g., EIA's NG processing facilities.

L460: I think this is not a proper citation. The authors should cite the DOGGR annual report instead of online GIS datasets for this purpose. I see some confusion between the dataset and the report here and there.

L484: grammar error.

L521 – L523: I think the majority of the emission sources described so fat are "point" sources. Because the authors identified the boundary of each facilities, it does not mean they are area sources. Area sources should be much broad and sometimes, unidentified sources. Although some sources may benefit from identifying the spatial extent, I don't see how useful the spatial extent for point-scale facilities (on a map) would be. One of the source sector that can really benefit from spatial delineation of

the facility boundary is the dairy section. Unfortunately, however, the authors do not provide this although they focus a lot on facilities that, in my opinion, do not require such information on spatial extent. I'd like to hear why it is important to figure out spatial extent for point-scale facilities. Is it helpful to perform atmospheric inversion for which a typical (even state-of-the-art) transport model can be run at the kilometer scale to capture underlying processes without significant errors. For airborne sampling planning, a simple point representation should be sufficient.

L538: The sentence is not clear.

L680: As commented above, please explain why this is a major advance. Please also remove "true" here. The problem here is that the process of making polygons require a tremendous amount of manual work, as stated in many places of this manuscript. I am not sure how the polygon features will be updated in the future, in larger applications for the state or other countries as the authors claim that theoretically this method can be applied in any regions; it can be done but the efficiency is in question. For some source sectors like dairy farms, I see the utility of polygons; they are large and sometimes located across multiple pixels on gridded maps. Just using GIS techniques for making polygons cannot be regarded as a scientific advancement that can be useful to the scientific community. Working on inverse problems for a long time, I don't see such a huge benefit from this polygon feature relative to the amount of the efforts (manual work) and potential and/or unidentified errors.

L694 – 696: Simple point-scale identification should be enough even for fugitive emission sources, in particular when gridded to kilometer or sub-kilometer scales (although sub-kilometer- scale simulations are not practical for most applications). Even for use with mobile surveys, a simple representation is enough when overlaid with Google (or other similar) maps.

L696 – 699: This is really strange: 1) where is the comparison result?; 2) Vista-LA has the result as "emissions"? The answer is no; and 3) when there is no emission

product from Vista, how can it be compared with other gridded emissions to reach this conclusion? Without ~10 km aggregation for all three (CALGEM, Vista, and EPA; because CALGEM and EPA maps are in ~ 10 km), the authors should not conclude like this.

L838 – 847: The authors are giving too much hope for automated feature extraction. The machine learning algorithm, in general, relies on cross-validation techniques and other simple statistics (e.g., mean absolute error) to validate the classification or regression results. If you look at the result of any cross validation (typically k-fold cross validation), it is not perfect; at a certain threshold point (e.g., 80% match between predictions and observations), you have to stop. This means it is associated with a lot of uncertainties due to limited training datasets, imperfect algorithms, etc. Without using complex techniques (e.g., bootstrapping, very computationally expensive) a typical machine learning algorithm does not provide uncertainty estimates (e.g., error for regressing fit). The authors need to state that there exist equal or even more uncertainties in the machine learning approach unless they can provide an example here. If you have raw data, using raw data may reduce uncertainty rather than using a machine learning technique.

Figure 5: I hope that the authors can explain what is the benefit of delineating spatial boundaries of landfills compared with the existing spatial inventory (e.g., Maasakkers et al., 2016) where the location is a simple point. The spatial resolution of gridded inventories (to be used for atmospheric inversions) need not be in meter scales because transport model cannot be run in meter scales. Then, I think a simple representation should be enough for clearly defined facilities like landfills. If you want to look at the boundary, viewing it over a Google (or similar) map should be fine. If the authors disagree, please explain why.

---

## Author Comment (AC1) · 25 Oct 2017

***Author's response to interactive comments on:***

**"Vista-LA: Mapping methane emitting infrastructure in the Los Angeles megacity"**

**by V. Carranza et al.**

The authors would like to thank anonymous reviewer #1 for detailed and thoughtful comments on the manuscript. Below we include responses to each comment. Our response is structured in the following format: (1) Referee comment, (2) Author's response, (3) Manuscript (MS) changes. Our responses and MS changes are highlighted in *blue* text. All changes to manuscript text were also tracked and highlighted using "track changes". The page and line numbers in the author's response and MS changes refer to page and line numbers in the "track changes" version of the manuscript.

Response to Anonymous Referee #1: *Interactive comment*

()

1. **Referee comment**: In general the manuscript describes an important GIS dataset for assisting with the improvement of methane emission estimates in the South Coast air basin. The authors make a strong case for why the methods described will be useful in other areas, especially in California, but even nationally and internationally. At this stage, the dataset does not include emissions themselves; but, the attributes data do include some activity information that will be useful in making bottom-up emission estimates in the future. A table that summarizes, for each major source type, the level of completeness of the activity data included in the dataset would be a useful addition to this manuscript. Are all, most, some, or none of the needed activity data included in the dataset for each of the major source categories included?

   a. ***Author response:*** This comment is very useful for describing our dataset. Although at this stage, the dataset does not include emissions estimates, a future goal of the data product is to combine both top-down (observation based) and bottom-up (activity based) emissions estimates, along with information provided via state and national reporting programs, to improve methane emissions estimates for the South Coast Air Basin.

      The referee requested inclusion of a table that summarizes, for each major source type, the level of completeness of the activity data included in the Vista-LA dataset. We have constructed this table and added it to the Supplementary Information. The attributes data for the Vista dataset do include some activity information that will be useful in making bottom-up emission estimates in the future. The amount of information currently included is indicated in Table S1 using a qualitative assessment ("none", "some" or "all"), as suggested by the reviewer.

   b. ***MS changes:*** Please review the added Supplementary Information; the new text includes Table S1 that that summarizes, for each major source type, the level of completeness of the activity data included in the Vista-LA dataset.

2. **Referee comment**: Lines 39-40: "Recent studies have shown that mitigating CH4 emissions yields large near-term climate benefits due to CH4's relatively short atmospheric lifetime (Dlugokencky et al., 2011)." Suggested clarification: Methane yields large near-term benefits do to its short lifetime AND its high GWP.

   a. *Author response:* We included this suggestion in the text in the Introduction.

   b. *MS changes:* Page 3, lines 37-39: "Recent studies have shown that mitigating $CH_4$ emissions yields large near-term climate benefits due to $CH_4$'s relatively short atmospheric lifetime **and high global warming potential** (Dlugokencky et al., 2011)."

3. **Referee comment**: Several of the statements in the Introduction section, added to support the utility of the dataset, appear to be in conflict: a) Urban areas are globally significant sources of methane (line 57); b) Urban methane is mostly from fossil fuel sources (line 72); c) By far, most methane in California is from livestock and waste (Figure 1). Please clarify the text in the Introduction section to explain how statements a) and b) do not conflict with information presented in Figure 1.

   a. *Author response:* Thank you for pointing out the logical disconnect here. We have amended the text to point out: (1) the discrepancies between inventories and atmospheric observations for $CH_4$ emissions in cities, and (2) the differences between statewide and city-scale $CH_4$ emission sources:

   b. *MS changes:* We have clarified the text as follows to address this comment:

      i. (a) Page 3, lines 58-59: The comment that "Urban areas are globally significant sources of $CH_4$ emissions" is correct and has not been changed. We added to the statement, "**however, correct quantification and source attribution at the scale of individual cities is highly uncertain**" to clarify point 1 above.

      ii. (b) The text stating that "Urban methane is mostly from fossil fuel sources" has been removed to avoid confusion.

      iii. (c) Figure 1: Correct, the largest sources of methane in the state of California are from livestock and waste. By contrast, fossil fuel emissions appear to be more important for some urban areas such as Los Angeles. We added a statement to clarify point 2 for cities in general, "**Official $CH_4$ emission inventories made using bottom-up approaches (e.g., Intergovernmental Panel on Climate Change (IPCC), 2006) are often created for policy and planning purposes at the state and national level (CARB, 2016; EPA, 2016); however, $CH_4$ sources in cities often differ substantially because of the high density of fossil fuel usage and relative lack of agricultural activities**. " (page 4, lines 63-66). We also added the following sentence to the caption of Figure 1: "**Note that while Livestock and Waste are the most significant sources of $CH_4$ in the state of California, atmospheric $CH_4$ in the Los Angeles urban**

**landscape is dominated by CH$_4$ hotspots from fossil fuel-derived sources (e.g., Hopkins et al., 2016b).”**

4. **Referee comment**: In the Introduction section, it would be helpful to mention the CALGEM dataset where other relevant datasets are discussed since CALGEM is discussed in the Discussion section (line 698).

   a. *Author response:* A good suggestion-- we have now included text that mentions the CALGEM and EPA inventories in the introduction.

   b. *MS changes:* Page 4, lines 77-79: “**Recent efforts have been made to spatially disaggregate these emissions by sector for California and United States inventories, resulting in 0.1º x 0.1º gridded CH$_4$ emissions products that coarsely represent the city scale (CALGEM: Jeong et al., 2013; EPA: Maasakkers et al., 2016).”**

5. **Referee comment**: Lines 714-715: “…rice cultivation and coal mining were the only source types contributing >1% of total emissions that were not included.” Statements here were confusing to me. Is the “total” referred here the US total? Please clarify.

   a. *Author response:* Also a good suggestion. The sentence was modified for consistency with Figure 1.

   b. *MS changes:* Page 32, lines 766-768: “Although rice cultivation and coal mining contribute >1% to total U.S. methane emissions, these are not significant sources of methane in SoCAB and were excluded from the Vista-LA database.”

6. **Referee comment**: Line 206 and line 239 mark subsections “Data processing and validation:” and “Limitations.” It is unclear why in other parts of the manuscript these subsections are combined. Recommend: Combine these subsections or separate the equivalent subsections to maintain parallel structure with the manuscript.

   a. *Author response:* The subsections were combined as “Data processing, validation, and limitations” to maintain consistent structure throughout the manuscript.

   b. *MS changes:* See manuscript page 9, line 219.

7. **Referee comment**: Line 325: “…these data under IPCC Level 3-1B2)” seems to be missing text.

   a. *Author response:* This was a typo.

   b. *MS changes:* A period was added to the end of the sentence.

8. **Referee comment**: Line 493: “data was” → “data were”.

a. ***Author response:*** The plural form "data were" was used changed in all relevant instances.

b. ***MS changes:*** See manuscript.

---

## Author Comment (AC2) · 25 Oct 2017

***Author's response to interactive comments on:***

**"Vista-LA: Mapping methane emitting infrastructure in the Los Angeles megacity"**

**by V. Carranza et al.**

The authors would like to thank anonymous reviewer #2 for detailed and thoughtful comments on the manuscript. Below we include responses to each comment. Our response is structured in the following format: (1) Referee comment, (2) Author's response, (3) Manuscript (MS) changes. Our responses and MS changes are highlighted in *blue* text. All changes to manuscript text were also tracked and highlighted using "track changes". The page and line numbers in the author's response and MS changes refer to page and line numbers in the "track changes" version of the manuscript.

Response to Anonymous Referee #2: *Interactive comment*

()

1. **Referee comments:**
   - General comments: The work presented here identifies potential sources of methane emission in South Coast Air Basin of California. I believe this is the first of its kind reported in terms of details on the locations of individual facilities and additional information attached to the spatial information. I think that the authors did a tremendous amount of work and the work deserves publication after addressing the comments.

   - **Comment 1:** I appreciate detailed information on how to process the data (e.g., geo-referencing) but the paper is too long. The information is useful and I recommend the authors move some of them to the supplemental.
   - **Comment 3b:** The results section is somewhat boring because it just lists datasets, not something meaningful to be absorbed.

     a. ***Author response:*** We chose to submit the Vista-LA paper to ESSD because much of the information on the data sources, products, and their validation is highly technical and the paper may be perceived as "too long" or "boring" by some, particularly readers used to journals oriented towards science results rather than data products. Nevertheless, this information is critical to the fidelity of the database and separating it into a Supplemental Information appendix would not accurately represent the provenance of Vista-LA.

     b. ***MS changes:*** None. We wish to keep the database details as part of the main paper.

   - **Comment 2:** I would suggest that the authors clearly state why the spatial extent (i.e., polygon delineation) is useful. As commented below, I am not convinced that the polygon is better than a simple representation as a point. This work will be useful for inversions but I don't see enough benefits from polygons, except for a few sectors

such as dairy farms. Polygon work is associated with errors and makes it difficult for the authors to update the database in the future; the authors already used a significant amount of time in manual work.

c. ***Author response:*** While our primary interest is in point sources, as Reviewer #2 notes, we have provided polygons in the Vista-LA database to expedite attribution since the area associated with a given facility, landfill, dairy, etc. is the ownership unit. Additionally, there may be many different point source classes within a given polygon – for example wellheads, storage tanks, compressors, or pipelines – and having the polygon/operator identity in our classification hierarchy provides critical information on traits such as behavior, processes, and facility history. We acknowledge that there is additional work associated with assembling and validating polygons, but feel that the additional information in the database offsets this extra cost.

d. ***MS changes:*** New text added in page 7, lines 162-167: "While point sources are our primary interest, we have provided polygons in Vista-LA to expedite attribution since the area associated with a given facility, landfill, dairy, etc. is the ownership unit. Additionally, there may be many different source classes within a given polygon − for example wellheads, storage tanks, compressors, or pipelines − and having the polygon/operator identity in our classification hierarchy provides critical information on traits such as behavior, processes, and facility history."

- **Comment 3a:** I realized that this work is not about mapping "emissions" when I expect some results on emission mapping. So the authors need to clarify this at the beginning.

  e. ***Author response:*** The text explicitly states that the current Vista-LA database is only a GIS representation of known or expected $CH_4$ sources and not a complete emissions inventory:
    i. (Page 1, lines 25-26): "The final database, Vista-Los Angeles (Vista-LA), is presented as maps of infrastructure known or expected to emit $CH_4$."
    ii. (Page 2, lines 29-31): "This study represents a first step towards developing an accurate, spatially-resolved methane flux estimate for point sources in SoCAB."
    iii. (Page 5, lines 102-105): "Vista-LA consists of detailed spatial maps for facilities and infrastructure in the SoCAB that are known or expected sources of $CH_4$ emissions, representing a first step towards developing an urban-scale $CH_4$ emissions gridded inventory for the SoCAB."
    iv. (Page 38, lines 929-932): "Vista-LA adopts a GIS-based approach to map known or potential $CH_4$ emissions sources in dense-mixed-land use areas of the South Coast Air Basin, which includes the LA Megacity. Characterizing $CH_4$ emissions at the urban scale is incredibly complex, as there exist thousands of structures known to be associated with $CH_4$ emissions. Vista-LA successfully identifies 33,554 potential $CH_4$ emitters from three IPCC sectors: Energy, Agriculture, and Waste."

     f.   *MS changes:* None.

- **Comment 4:** There are many figures, but the authors do not link the text with figures, only providing a minimum description about the figures.

     g.   *Author response:* Thank you for noting this deficiency.

     h.   *MS changes:* New text has been added throughout the paper to call out individual figures and provide more complete context of the information they contribute to the paper( e.g., please see "track changes" version of MS: page 11, line 259; page 13, line 311; page 15, lines 360-361; page 16, lines 386-387)

2. **Referee comment**: L20: essentially SoCAB?

     a.   *Author response:* Yes, we focus on the entire South Coast Air Basin in Vista-LA since it contains the majority of LA Megacity GHG emissions. The text explains the spatial extent of Vista-LA in page 6, lines 118-120: "The spatial domain for the Vista-LA database is SoCAB, the air-shed for the greater Los Angeles urban extent, including all of Orange County, the urbanized parts of Los Angeles, Riverside, and San Bernardino Counties." In addition, we made changes to the main text to further clarify this point.

     b.   *MS changes:*

          i.   Page 1, lines 16-18: "Here, we present Vista, a Geographic Information System (GIS)-based approach to map potential methane emissions sources **in the South Coast Air Basin (SoCAB) that encompasses greater Los Angeles,** an area with a dense, complex mixture of methane sources."

         ii.   We provided an explanation of the spatial extent earlier in text:

              1.   Page 5, lines 95-99: "Here, we present Vista, a Geographic Information System (GIS)-based approach to map potential methane emissions sources in the South Coast Air Basin (SoCAB), **which includes all of Orange County, and the non-desert regions of Los Angeles County, Riverside County, and San Bernardino County**. Our primary goal is to improve understanding of $CH_4$ emissions at urban scales with complex mixtures of sources, exemplified by the LA Megacity **within SoCAB**."

3. **Referee comment**: L33: globally or California or somewhere else?

     a.   *Author response:* Vista-LA provides the GIS data associated with known or potential $CH_4$ sources in the South Coast Air Basin. It does not reconcile bottom-up and top-down emissions, but has the potential to address this issue in the future. We added text to make this clear.

      b. *MS changes:* Page 2, lines 29-31: "This study represents a first step towards developing an accurate, spatially-resolved methane flux estimate for point sources in SoCAB, with the potential to address discrepancies between bottom-up and top-down methane emissions accounting **in this region**."

4. **Referee comment**: L41: very vague. "at scales relevant to actionable emissions reduction efforts", Which scale?

      a. *Author response:* We replaced the vague text with a more specific description.

      b. *MS changes:* Page 3, line 40: "**policy-relevant spatial** scales **(e.g., cities to nations).**"

5. **Referee comment**: L48: any reference for the urban policy scale? Why the entire city? In SoCAB, there are so many small cities. Within those small cities, policies are different from meter scales? Activities vary significantly place to place, but I don't believe such a large variation in urban policies, in particular for GHG control.

      a. *Author response:* The reviewer is correct that the SoCAB domain includes many small cities in addition to Los Angeles, that are not easily distinguishable from Los Angeles due to geography (e.g., with respect to atmospheric measurements, economic activity, etc.). We are not suggesting that these cities differ from one another. However, fine-scale information is needed because there is a good deal of spatial variability in emissions at this level. We changed the sentence to remove reference to the urban policy scale, and added a new sentence that describes spatial patterns of emissions.

      b. *MS changes:* Removed "align with urban policy and planning (typically 10s to 100s of meters)," and replaced with "**reflect their variability across the landscape**" on page 3, line 47. Added new sentence, page 3, lines 52-55: "**Studies of spatial patterns of urban CH$_4$ demonstrate fine-scale variability, with CH$_4$ concentrated in hotspots compared to more evenly dispersed CO$_2$ (Hopkins et al., 2016b). This pattern reflects how** the sources of CH$_4$ differ…"

6. **Referee comment**: L52-54: This needs more explanations. Why do we need such a fine-scale emission map? Because the sources of urban CH4 are mostly co-located with those of CO2? Here, we need to be clear. For simple inventory purposes, we may need really fine-scale maps. But as a prior model in atmospheric inversion, do we really need meter-scale maps when the transport model cannot really resolve at such a fine scale? Please add more comments. Otherwise, it sounds like "we just developed a fine-scale product because it is better."

      a. *Author response:* We appreciate the opportunity to clarify the need for fine scale maps of methane—see response to previous referee comment (above). We made small changes to language to point to CH$_4$ emission maps rather than products to further reduce confusion.

b. *MS changes:* See response to previous referee comment (page 3, line 47, 52-55), and changes on page 3, line 51: "**Similar** CH$_4$ emission **maps** with spatial information…"

7. **Referee comment**: L54 – 56: agreed.

    a. *Author response:* Left statement as is.

    b. *MS changes:* No changes.

8. **Referee comment**: L58: we know the location of a lot of sources for urban areas relative to rural areas. What we don't know is the activity levels and emissions.

    a. *Author response:* In response to Reviewer #1, we amended this statement to clarify our point regarding the uncertainty of emissions in urban areas. Please see response to Reviewer #1 for further explanations.

    b. *MS changes:* Page 3, lines 58-59: "Urban areas are globally significant sources of CH$_4$ emissions; **however, correct quantification and source attribution at the scale of individual cities is highly uncertain**."

9. **Referee comment**: L63: reference? I think at least Wunch et al., 2009 mentioned this limitation.

    a. *Author response:* Thank you for the suggestion. The reviewer is correct that Wunch et al. (2009) were the first study to compare EDGAR CH$_4$ to atmospheric observations in SoCAB—we added this reference.

    b. *MS changes:* Page 4, line 60-62: "(e.g., EDGAR v4.2 European Commission Joint Research Centre, 2010; Olivier and Peters, 2005) are limited in their usefulness for estimating emissions at the scale of a city or air basin, **as demonstrated for SoCAB by Wunch et al. (2009).**"

10. **Referee comment**: L72 – 73: What about landfill. In SoCAB, fossil-fuel sources cannot necessarily be "dominant" given the uncertainty of recent top-down studies in the region. Furthermore, depending on the definition of the hot spot, hot spots from fossil sources alone cannot be the dominant source in SoCAB. I feel that the authors touch lots of things related to urban CH4 emissions, but I am not sure that the authors are making effective arguments towards the goal of this new study. Literature review is not only scattered here and there but also somewhat inaccurate. I would recommend that the authors revise the introduction section with more clarity and accurate statements.

    a. *Author response:* We have revised the introduction section in response to Reviewer #1 to provide more context to our point that fossil fuel emissions appear to be more dominant in the Los Angeles urban area. We also revised the introduction section with an effort to provide more clarity. Please refer to our response to Reviewer #1 for further explanations.

      **b.** *MS changes:*

          **i.** The text stating that "Atmospheric $CH_4$ in the urban landscape is dominated by $CH_4$ hotspots that primarily come from fossil fuel-derived sources (e.g., Hopkins et al., 2016b)." was removed in the main introduction.

          **ii.** We also added the following sentence to the caption of Figure 1: "**Note that while Livestock and Waste are the most significant sources of $CH_4$ in the state of California, atmospheric $CH_4$ in the Los Angeles urban landscape is dominated by $CH_4$ hotspots from fossil fuel-derived sources (e.g., Hopkins et al., 2016b)."**

**11. Referee comment**: L89: Did the authors clearly lay out the shortcomings of current urban inventories. Which urban inventory? Here is the problem. The authors try to deal with the "general" urban CH4. Please focus on SoCAB. The authors don't even mention which inventory is available for SoCAB. There are several spatially explicit emission maps that include California and SoCAB. What are the shortcomings of those? The authors should spare the introduction section in describing weaknesses with current inventories (CARB, EDGAR, EPA spatial (ES&T, 2016), CALGEM, etc.) and the contribution this work can make over them.

      **a.** *Author response:* We include text in page 4, lines 59-74 and page 4, lines 77-86 (below) that further explains the shortcomings of current urban inventories, including those that show discrepancies between bottom-up inventories and top-down observations in Los Angeles. We explicitly state that $CH_4$ inventories of the greater Los Angeles region underestimate emissions by 40-50% according to estimates from atmospheric observations.

      **b.** *MS changes:*

          i. Page 4, lines 59-74: "Global emissions inventories based on nightlights and/or population scaling methods (e.g., EDGAR v4.2 European Commission Joint Research Centre, 2010; Olivier and Peters, 2005) are limited in their usefulness for estimating emissions at the scale of a city or air basin, as demonstrated for SoCAB by Wunch et al. (2009). Official $CH_4$ emission inventories made using bottom-up approaches (e.g., Intergovernmental Panel on Climate Change (IPCC), 2006) **are often created for policy and planning purposes at the state and national level (CARB, 2016; EPA, 2016); however, $CH_4$ sources in cities often differ substantially because of the high density of fossil fuel usage and relative lack of agricultural activities. Yet bottom-up inventories have still been shown to underestimate $CH_4$ emissions, contain inaccurate information about the distribution of emissions sources, or have incorrect source apportionment compared to atmospheric observations.** Such discrepancies have been observed in many North American and European cities, including the greater Los Angeles (LA)

region, **where CH$_4$ inventories consistently underestimate emissions by 40-50% based on estimates from atmospheric observations** (Hopkins et al., 2016b; Hsu et al., 2009; Townsend-Small et al., 2012; Wennberg et al., 2012; Wong et al., 2016, 2015; Wunch et al., 2009), and in other cities such as Boston (McKain et al., 2015), Indianapolis (Cambaliza et al., 2015), Florence (Gioli et al., 2012), London (Helfter et al., 2016), and San Francisco (Jeong et al., 2017)."

ii. Page 4, lines 77-86:" Recent efforts have been made to spatially disaggregate these emissions by sector for California and United States inventories, resulting in 0.1° x 0.1° gridded CH$_4$ emissions products that coarsely represent the city scale (CALGEM: Jeong et al., 2013; Maasakkers et al., 2016); **however, these scales are still too coarse for interpreting new fine-scale observations. Inaccuracies and coarse spatial information in city-scale CH$_4$ emission inventories pose a direct obstacle to city mitigation plans. One hypothesis for the discrepancy between CH$_4$ observations and inventories in cities is that fugitive emissions, particularly from natural gas systems, are currently underrepresented in inventories. A related hypothesis suggests that this discrepancy stems from undercounting disproportionately large CH$_4$ "super-emitters" such as those that have been shown to occur in natural gas systems (Brandt et al., 2014)**."

12. **Referee comment**: L93: For those outside California, the use of LA Megacity is confusing since the authors introduced SoCAB. For inventory purposes, SoCAB should be preferable because CARB uses this air basin for regulatory purposes.

   a. *Author response:* The text clearly states that the LA Megacity is part of SoCAB in the abstract (page 1, lines 17-18), introduction (page 5, line 99), and that SoCAB is an ideal testbed (page 5, lines 106-110): "SoCAB is an ideal testbed due to the density of sources and availability of observations from the LA Megacity Carbon Project (https://megacities.jpl.nasa.gov/portal/) tower network (Newman et al., 2016; Verhulst et al., 2017), the California Laboratory for Atmospheric Remote Sensing (CLARS) (Wong et al., 2016, 2015), and a total column carbon observing network site (Wunch et al., 2009)."  See also our response #2 above.

   b. *MS changes:* We also added text in the conclusion that reiterates that the LA Megacity is part of SoCAB:

      i. Page 38, lines 929-930: "Vista-LA adopts a GIS-based approach to map known or potential CH$_4$ emissions sources in dense-mixed-land use areas of the South Coast Air Basin, **which includes the LA Megacity**."

13. **Referee comment**: L100: What are potential source vs. facilities/infrastructure sources?

a. ***Author response:*** Potential sources consist of facilities/infrastructure sources that may be contributing to methane emissions in the South Coast Air Basin, but have not yet been measured. We have amended the text in the introduction that better describes what the Vista-LA data product represents.

b. ***MS changes:*** Page 5, lines 102-105: "Vista-LA consists of detailed **spatial maps for facilities and infrastructure in the SoCAB that are known or expected sources of CH$_4$ emissions,** representing a first step towards developing an urban-scale CH$_4$ emissions gridded inventory for the SoCAB."

14. **Referee comment**: L193: I am curious about the portion of transportation in SoCAB. It would be useful to look at the transportation emission for SoCAB from Maasakkers et. al. 2016. Although the portion of transportation relative to the total may be small, transportation seems to be important in this high-resolution maps, in particular in this highly urbanized region.

a. ***Author response:*** We thank the reviewer for this question, which we addressed in page 32, lines 775-781: "We omitted several categories that might have important contributions to CH$_4$ emissions in SoCAB, such as transportation. Although transportation produces ~1% of California inventoried CH$_4$ emissions (and <0.3% of national emissions; EPA, 2016), it likely comprises a greater fraction of SoCAB emissions given the greater density of traffic in the region. We have chosen not to include a spatial layer for transportation in this version of Vista; we view Vista primarily as a tool for attribution of large fugitive CH$_4$ emission sources, and there is no evidence for this type of emission from conventionally fueled vehicles."

   In brief, we decided to omit Transportation emissions (1A3) in this initial version of Vista-LA since they are estimated to account for ~1% of total statewide emissions and should be negligible (Fig. 1). As the reviewer correctly points out in Comment 28, this version of Vista-LA is not an emissions inventory. Hence, it is not valid to compare Vista-LA to the Maasakkers et al. emission product.

b. ***MS changes:*** None.

15. **Referee comment**: L212: I wonder if EIA data for refinery provide spatial information (i.e., lon/lat), probably only zip code. If only zip code, then this sentence is not really correct. EPA mandatory reporting (GHGRP) may provide exact locations.

a. ***Author response:*** Yes, the raw EIA shapefile already provides latitude and longitude information.

b. ***MS changes:*** None.

16. **Referee comment**: L217 – 224: It should be useful to check with EPA's mandatory GHG reporting system.

a. *Author response:* EIA was verified with GHGRP. They both contained information for a total of 9 refineries for SoCAB.

b. *MS changes:* Page 10, Lines 222-224: "SCAG and **the U.S. Environmental Protection Agency's Facility Level Information on GHG online reporting Tool (EPA FLIGHT)** was used to verify that there were no missing petroleum refineries from EIA."

17. **Referee comment**: L228: Please confirm that the EIA dataset has exact point locations. Also, crosschecking with Facility Registry Service (FRS) should be useful

   a. *Author response:* EIA has point locations. There is no need to cross-check since there are only 9 refineries which were easily verified using aerial imagery. FRS only gives information on 2 refineries in SoCAB.
   b. *MS changes:* None.

18. **Referee comment**: L252: reason for claiming "The Vista-LA power plant dataset provides accurate location and extent data"?

   a. *Author response:* We state, "The Vista-LA power plant dataset provides accurate location and extent data" to illustrate and emphasize that this dataset has undergone significant QA/QC and is by far more spatially resolved and validated than any readily available public sources to date (EIA, FFDAS, ODIAC). It not only contains location and metadata information, but also contains an accurate outline (extent) of each power plant facility, which is critical when attributing emissions to form a bottom-up grid, and again won't be found in any readily available public dataset.
   b. *MS changes:* None.

19. **Referee comment**: L372: The authors may want to verify the compressor stations in SoCAB comparing with those from the California Energy Commission.

   a. *Author response:* At the time this manuscript was submitted we did not have permission to share data from the California Energy Commission. Since then we have gained permission and will include the location of natural gas compressor stations as static points in Figures 2 and 3. However, we had to sign an NDA, so we cannot add this data to the Vista-LA product.

   b. *MS changes:* Added text to page 16, lines 392-394: "The natural gas compressor station (IPCC – 1B2) dataset was obtained using the U.S. Environmental Protection Agency's Facility Level Information on GHG online reporting Tool (EPA FLIGHT) and California Energy Commission (CEC).

20. **Referee comment**: L396: Why did the authors use the EIA pipelines instead of those of CEC? Due to security concerns although the authors do not state it (only for NPMS).

a. ***Author response:*** This is correct, please see page 17, lines 412-413: "Due to security concerns, the CEC dataset is only shown as static representations in Figures 2 and 3."

b. ***MS changes:*** None.

21. **Referee comment**: L410: Don't need repeat this geo-referencing unless there was a need to manually geolocate facilities, e.g., EIA's NG processing facilities.

a. ***Author response:*** Thank you for the suggestion. We have removed the explanation about geo-referencing in this context.

b. ***MS changes:*** Removed "Both datasets were first georeferenced" on page 19.

22. **Referee comment**: L460: I think this is not a proper citation. The authors should cite the DOGGR annual report instead of online GIS datasets for this purpose. I see some confusion between the dataset and the report here and there.

a. ***Author response:*** The DOGGR Annual Report provides information and statistics about the annual oil and gas production in California, but does not provide information about the accuracy, scale, origin, and completeness of GIS data of oil and gas wells in California. We amended the text to clarify that we are specifically referring to GIS data. In this case, we think that the reference to the online GIS datasets is more appropriate than the DOGGR Annual Report.-

b. ***MS changes:*** Page 20, lines 481-482: "According to DOGGR, **GIS data of oil and gas wells** varies in accuracy, scale, origin and completeness (DOGGR, 2016)".

23. **Referee comment**: L484: grammar error.

a. ***Author response:*** Agreed. The grammar error was fixed.

b. ***MS changes:*** Page 21, lines 512-514: "Emissions from manure depend **on** the type of management practices employed by the farm or facility (Kaffka et al., 2016)."

24. **-Referee comment**: L521 – L523: I think the majority of the emission sources described so fat are "point" sources. Because the authors identified the boundary of each facilities, it does not mean they are area sources. Area sources should be much broad and sometimes, unidentified sources. Although some sources may benefit from identifying the spatial extent, I don't see how useful the spatial extent for point-scale facilities (on a map) would be. One of the source sector that can really benefit from spatial delineation of the facility boundary is the dairy section. Unfortunately, however, the authors do not provide this although they focus a lot on facilities that, in my opinion, do not require such information on spatial extent. I'd like to hear why it is important to figure out spatial extent for point-scale facilities. Is it helpful to perform atmospheric inversion for which a typical (even state-of-the-art) transport model can be run at the kilometer

scale to capture underlying processes without significant errors. For airborne sampling planning, a simple point representation should be sufficient.

    a. *__Author response:__* We thank the reviewer for pointing out the potential for misunderstanding with regards to polygons. We are not asserting that these polygons represent "area sources" in all cases—while some are true area sources (e.g., landfills, dairies), others are better characterized as a collection of point sources (e.g., oil refineries, natural gas storage fields). We clarified the text to better reflect the definition of polygon sources vs. area sources, and to explain why they are important.

    b. *__MS changes:__* Page 30, lines 720-735: "**Nine source types are represented as polygons; some represent "area" sources, such as landfills, while others represent a combination of point sources within a facility, such as oil refineries. We have chosen to represent both types of sources with polygons, since Vista-LA is a facility-level database; at present we do not have sub-facility scale information. These polygons depict the true spatial extent of each facility, enabling improved, and potentially automated attribution of methane plumes observed in airborne imaging or mobile *in situ* surveys. In survey data, observed methane plumes may not be close to point representations of the address of a facility, and hence may not be easily attributed to the emitting facility, particularly in complex surroundings with many closely located facilities such as SoCAB (e.g., El Segundo or Port of Los Angeles/Long Beach areas). Polygon representations enable an automated workflow where the geolocation of a plume detection is attributed to an entity in the Vista-LA database of sources using spatial intersection without any manual work by a human operator. Source attribution to the facility level is also useful because emissions data are often reported as an aggregate value representing the facility level (e.g., CARB's Mandatory GHG Reporting Regulation).** The remaining two sources currently represented by points − dairies and anaerobic lagoons − require future work to accurately describe their spatial extents."

25. **Referee comment**: L538: The sentence is not clear.

    a. *__Author response:__* Agreed. We amended the sentence to make it clear.

    b. *__MS changes:__* Page 24, lines 568-569: "Therefore, we created a preliminary geospatial dataset of **anaerobic lagoons based on GIS data of dairies and cattle farms**."

26. **Referee comment**: L680: As commented above, please explain why this is a major advance. Please also remove "true" here. The problem here is that the process of making polygons require a tremendous amount of manual work, as stated in many places of this manuscript. I am not sure how the polygon features will be updated in the future, in larger applications for the state or other countries as the authors claim that theoretically this method can be applied in any regions; it can be done but the efficiency is in question. For some source

sectors like dairy farms, I see the utility of polygons; they are large and sometimes located across multiple pixels on gridded maps. Just using GIS techniques for making polygons cannot be regarded as a scientific advancement that can be useful to the scientific community. Working on inverse problems for a long time, I don't see such a huge benefit from this polygon feature relative to the amount of the efforts (manual work) and potential and/or unidentified errors.

    a. *Author response:* The reviewer is correct that the polygons do not represent a scientific advance, but rather an advance in the sophistication of available datasets that enables automation of further analyses. The reviewer is correct that creating polygons requires a tremendous amount of manual work, but we foresee that this advance work will allow us to remove the human from the loop for future source attribution studies. At least for SoCAB, it is unlikely that the polygon features would need to be updated frequently since most facilities in this developed regions are not often changing their spatial extents. However, we acknowledge the challenge this would pose in rapidly developing regions. Again, we did not intend the polygons to improve inversions, but rather are primarily a tool for source attribution of emissions from surveys.

    b. *MS changes:* Please see MS changes made in response to referee comment #24 (page 30, lines 720-735).

27. **Referee comment**: L694 – 696: Simple point-scale identification should be enough even for fugitive emission sources, in particular when gridded to kilometer or sub-kilometer scales (although sub-kilometer- scale simulations are not practical for most applications). Even for use with mobile surveys, a simple representation is enough when overlaid with Google (or other similar) maps.

    a. *Author response:* We respectfully disagree that point features are sufficient for source attribution in survey data. To provide an example, the El Segundo refinery is nearly 4 km$^2$ in area. However, if one were to draw a circle with a 1 km diameter near the western edge of this facility, there would be 2 power plants, 3 oil wells, and a wastewater treatment plant in addition to the refinery as possible methane sources. Hence any methane plumes observed in this <1 km$^2$ area could possibly be attributed to a number of different sources/sectors. By providing polygons, it becomes much clearer which facility the methane plume in fact belongs to.

    b. *MS changes:* Please see MS changes made in response to referee comment #24 (page 30, lines 720-735).

28. **Referee comment**: L696 – 699: This is really strange: 1) where is the comparison result?; 2) Vista-LA has the result as "emissions"? The answer is no; and 3) when there is no emission product from Vista, how can it be compared with other gridded emissions to reach this conclusion? Without ~10 km aggregation for all three (CALGEM, Vista, and EPA; because CALGEM and EPA maps are in ~ 10 km), the authors should not conclude like this.

a. *Author response:* Correct, this version of Vista-LA does not have an emissions result and, as such, we do not compare emissions between the existing inventories and Vista-LA. We amended the text that generally describes why the fine-scale spatial resolution of Vista-LA can be useful for capturing the nature of $CH_4$ hotspots in urban areas compared to more coarsely gridded products.

b. *MS changes:* Page 31, lines 747-450: "This **spatial structure** more closely matches the "hotspot" nature of atmospheric $CH_4$ that has been observed in SoCAB **at the scale of meters to kilometers** (Hopkins et al., 2016b) **than is represented by existing products that are too coarse to capture the fine-scale nature of $CH_4$ hotspots in urban areas.**"

29. **Referee comment**: L838 – 847: The authors are giving too much hope for automated feature extraction. The machine learning algorithm, in general, relies on cross-validation techniques and other simple statistics (e.g., mean absolute error) to validate the classification or re gression results. If you look at the result of any cross validation (typically k-fold cross validation), it is not perfect; at a certain threshold point (e.g., 80% match between predictions and observations), you have to stop. This means it is associated with a lot of uncertainties due to limited training datasets, imperfect algorithms, etc. Without using complex techniques (e.g., bootstrapping, very computationally expensive) a typ ical machine learning algorithm does not provide uncertainty estimates (e.g., error for regressing fit). The authors need to state that there exist equal or even more uncer tainties in the machine learning approach unless they can provide an example here. If you have raw data, using raw data may reduce uncertainty rather than using a machine learning technique.

a. *Author response:* We modified the text to convey the uncertainty that a machine learning technique may introduce. Although there is uncertainty with this technique, automated feature extraction has the potential to help expedite the process of locating infrastructures where GIS datasets currently do not exist (e.g. anaerobic lagoons from dairies) at a larger scale.

b. *MS changes:*

i. Removed the word "precisely" from page 37. Also removed the sentence describing the software on page 37.

ii. Add text to page 37, lines 905-909: "**It is important to note that a machine learning algorithm may introduce uncertainty more than the use of raw datasets, therefore efforts need to be made to quantify these uncertainty estimates. Even so, machine learning algorithms hold the potential to advance the process of identifying infrastructure of potential $CH_4$ sources at a larger scale.**"

30. **Referee comment:** Figure 5: I hope that the authors can explain what is the benefit of delineating spatial boundaries of landfills compared with the existing spatial inventory (e.g., Maasakkers et al., 2016) where the location is a simple point. The spatial resolution of

gridded inventories (to be used for atmospheric inversions) need not be in meter scales because transport model cannot be run in meter scales. Then, I think a simple representation should be enough for clearly defined facilities like landfills. If you want to look at the boundary, viewing it over a Google (or similar) map should be fine. If the authors disagree, please explain why.

a. ***Author response:*** Again, our primary intent of delineating the spatial boundaries of large facilities such as landfills with polygons is not to improve atmospheric inversions, but to assist with source attribution. Certainly it is easy to view in Google, however, it requires a human operator. The point of Vista-LA is that the human part of the work has already been done, providing a product that can hopefully be used in an automated way.

b. ***MS changes:*** Please see MS changes made in response to referee comment #24 (page 30, lines 720-735).